# Description of Lepiotaceous Fungal Species of the Genera *Chlorophyllum, Clarkeinda, Macrolepiota, Pseudolepiota,* and *Xanthagaricus,* from Laos and Thailand

**Phongeun Sysouphanthong** [1,2,3], **Naritsada Thongklang** [1,2,*], **Jian-Kui Liu** [4] and **Else C. Vellinga** [5]

1   Center of Excellence in Fungal Research, Mae Fah Luang University, Chiang Rai 57100, Thailand; laofungi@gmail.com
2   School of Science, Mae Fah Luang University, Chiang Rai 57100, Thailand
3   Ecology Division, Biotechnology and Ecology Institute, Ministry of Science and Technology, Vientiane P.O. Box 2279, Laos
4   School of Life Science and Technology, Center for Informational Biology, University of Electronic Science and Technology of China, Chengdu 611731, China; liujiankui@uestc.edu.cn
5   UC Herbarium, UC Berkeley, Berkeley, CA 94720-2465, USA; ecvellinga@comcast.net
*   Correspondence: naritsada.t@gmail.com; Tel.: +66-539-16996

**Abstract:** In our ongoing research on lepiotaceous taxa (Agaricaceae s.l.) in Laos and northern Thailand, we focus here on *Chlorophyllum, Clarkeinda, Macrolepiota, Pseudolepiota,* and *Xanthagaricus*. Collections were obtained from various habitats, including agricultural habitats, grasslands, and rainforests. A total of 12 taxa were examined and investigated. Of these 12, two are new for science; viz. *Xanthagaricus purpureosquamulosus* with brownish-grey to violet-brown squamules on a pale-violet to violet background; it shares the pileus color with *X. caeruleus* and *X. ianthinus*, but differs in other characters; and *Macrolepiota excelsa,* rather similar to *M. procera* but related to *M. detersa*. Two species, *Pseudolepiota zangmui* and *Xanthagaricus necopinatus* are recorded for the first time in Thailand. Four species of *Chlorophyllum* and a total of four species of *Macrolepiota* were found, viz., *C. demangei* and *C. hortense* with white basidiospores, *C. molybdites* and *C. globosum* with green basidiospores, *M. detersa, M. dolichaula,* the new *M. excelsa,* and *M. velosa*. Another rather common striking species is *Clarkeinda trachodes,* with yellow-green basidiospores. Each species is described in detail, with color photographs and line drawings. Phylogenetic analyses based on internal transcribed spacer (nrITS) region, the large subunit nuclear ribosomal (nrLSU) DNA and RNA polymerase II second largest subunit (*rpb2*) genes provide evidence for the placement of the species covered.

**Keywords:** Agaricaceae; biodiversity; new taxa; new record; taxonomy; phylogeny; Southeast Asia

## 1. Introduction

As part of ongoing survey into the mycological diversity of Laos and northern Thailand [1], the focus of this article is on *Macrolepiota* Singer, and some genera in the 'wider Agaricus clade' (also considered to be Agaricaceae in the strict sense [2–4], in particular, the genera *Chlorophyllum* Massee, *Clarkeinda* Kuntze, *Pseudolepiota* Z.W. Ge, and *Xanthagaricus* (Heinem.) Little Flower, Hosag. and T.K. Abraham. Earlier, we described two new genera in this clade, viz. *Coniolepiota* Vellinga and *Eriocybe* Vellinga [2] focused on the genus *Lepiota* (Pers.) Gray [5–8] and documented the rarely recorded species *Verrucospora vulgaris* Pegler [9].

*Chlorophyllum* Massee is widespread in tropical to temperate regions, often in anthropogenic landscapes, with a total of 19 species [10–12]. The species are agaricoid or secotioid, the latter occurring in steppes, deserts, and other dry habitats. Until 2003 *Chlorophyllum* was restricted to the green-spored agaricoid species, whereas the white-spored agaricoid taxa were included in *Macrolepiota* Singer, and the secotioid taxa were considered

part of *Endoptychum* Czern. or *Secotium* Kunze. Molecular-phylogenetic research clearly showed that *Chlorophyllum* includes white-spored taxa with a smooth stipe and with or without a simple germ pore, that *Chlorophyllum* and *Macrolepiota* s. str. are not closely related [2,13,14], and that the secotioid taxa are relatively recent morphological adaptations (e.g., Vellinga et al. 2003 [12,13,15]). A proposal to conserve the name *Chlorophyllum* against *Secotium* was published [16]. Only three *Chlorophyllum* species are listed in the checklist for Thailand [17]. Of these three, *Chl. molybdites* (G. Mey.) Massee seems to be common and widespread; *Chl. rhacodes* (Vittad.) Vellinga was reported from the northeast, while *Chl. hortense* (Murrill) Vellinga was only recorded from the southern part of Thailand [17–19]. However, these reports are based on morphology alone, and in this group of similar-looking species, it is better to have molecular and morphological evidence. The occurrence of *Chl. rhacodes* has not yet been confirmed in Thailand, but in this article, we report *Chl. globosum* (Mossebo) Vellinga with green basidiospores, and *Chl. demangei* (Pat.) Z.W. Ge and Zhu L. Yang with white basidiospores as additions to the Thai mycoflora.

*Clarkeinda trachodes* (Berk.) Singer, originally described from Sri Lanka as *Agaricus trachodes* Berk. [20], is commonly found in tropical regions of eastern and southern Asia; there are reports from Bangladesh [21], India [22,23], Malaysia [24], Indonesia, and southern China [25]. The species was also recorded from Thailand [2,17]. Here, we describe the Thai collections in detail.

*Macrolepiota* Singer is also widespread, and members of this genus can be found in various habitats, from grasslands to native forests. Till the early 2000s, the smooth-stiped species, now accommodated in *Chlorophyllum,* were considered part of *Macrolepiota*. However, *Macrolepiota* species differ from *Chlorophyllum* in the presence of a vesture on the stipe, resulting in a tiger pattern in the full-grown specimens, the trichodermal appearance of the pileus covering, and the presence of a real germ pore in the spores [13,26,27]. The two genera are not closely related, with *Chlorophyllum* in the *Agaricus* clade and *Macrolepiota* close to *Lepiota* [13]. Similar to *Chlorophyllum*, *Macrolepiota* also harbors secotioid taxa [28]. Seven *Macrolepiota* taxa have been reported from Thailand, viz. *M. africana* (R. Heim) Heinem, *M. dolichaula* (Berk. and Br.) Pegler and Rayner, *M. excoriata* (Schaeff.) Wasser, *M. gracilenta* (Krombh.) Wasser, *M. mastoidea* (Fr.) Singer, *M. procera* var. *procera* and *M. zeyheri* Heinem [17,18,29–31]. However, these records are based on morphology only; the presence of some of these species in Thailand has to be confirmed by molecular means.

The monotypic genus *Pseudolepiota* Z.W. Ge was recently described from southern China, based on *Ps. zangmui* Z.W. Ge [32]. This is one of the species superficially resembling the European species *Lepiota fuscovinacea* F.H. Møller and J.E. Lange with a cutis-like pileus covering and no clamp connections [2]. The species was found in two different locations in northern Thailand, expanding the known distribution considerably.

*Xanthagaricus* (Heinem.) Little Flower, Hosag. and T.K. Abraham is thus far only known from southern and eastern Asia and South Africa. It is a small genus characterized by small basidiomata, a squamulose pileus, white to pale-yellow basidiospores, and the absence of clamp connections. Heinemann and Little Flower [33] considered it a subgenus of *Hymenagaricus* Heinem., but recent phylogenetic research showed that it did not form a monophyletic group with *Hymenagaricus* species [34]. Two species were described from Thailand, viz., *X. thailandensis* J. Kumla, N. Suwannarach and S. Lumyong [35] and *X. siamensis* Yuan S. Liu and S. Lumyong [36], and here we describe a third species. We also record *X. necopinatus* Iqbal Hosen, T.H. Li, and G.M. Gates, described from Bangladesh, from Chiang Mai and Chiang Rai provinces in northern Thailand, expanding the known distribution of that species.

In this study, we aim to give descriptions of new species and report new records and extensive descriptions of some more common species for northern Thailand and Laos, based on morphological studies and comparisons and molecular phylogenetic analyses.

## 2. Materials and Methods

### 2.1. Sample Collection and Morphological Study

The specimens were collected during the wet seasons of 2007–2019 in Laos and Thailand. In the field, the forest type and habitat or substrate of mushrooms were recorded, and the samples were photographed. Macro-morphology, examined on fresh samples, included notes on shape, size, and color of the basidiomata, pileus, lamellae, stipe, annulus, context, and spore print following the glossary of Vellinga and Noordeloos [26] and Kornerup and Wanscher [37] for color annotations. Samples were dried on hot, dry air for 24 h and preserved in plastic bags with silica gel. Thai specimens were deposited in the herbarium at Mae Fah Luang University (MFLU) with duplicates for the 2007 collections made by Vellinga at UC; and Lao specimens were deposited in the National Herbarium of Laos (HNL).

The micromorphology was illustrated from dried material: basidiospores, basidia, and cheilocystidia from the lamellae and the structure of the pileus covering, using a drawing tube attached to an Olympus CX-41 research compound microscope. The original color of all characteristics was observed in water ($H_2O$) and 3–10% of potassium hydroxide (KOH), and chemical reactions were observed using Melzer's reagent, Cotton Blue, and Cresyl Blue; line drawings were made of material mounted in Congo Red in ammonia. A total of 25 basidiospores, 15 basidia, and cheilocystidia per collection were measured. Spores were measured in side-view. The notation [50/2/2] indicates that the measurements were made on 50 basidiospores in 2 samples in 2 collections; the size averages are given in the description, while quotient (Q) of length and width average was calculated to indicate the basidiospore shape. The technical terms used for the descriptions followed those of Vellinga and Noordeloos [26].

### 2.2. Molecular Study

DNA was extracted from dried herbarium collections, according to the instructions of the Biospin Fungus Genomic DNA Extraction Kit (Bioer Technology Co., Ltd., Hangzhou, China). Primers ITS1 and ITS4 were used for the nrITS1, 5.8S, and nrITS2 regions; primers LR0R and LR5 for large subunit region (LSU); primers fRPB2-6F and fRPB2-7R for polymerase II second largest subunit (*rpb2*) region; and PCR conditions followed common protocols [38,39]. The PCR amplified products were cleaned and sequenced by Shanghai Majorbio Bio-Pharm Technology Co., Ltd. (Shanghai, China). Sequences were edited, and contigs were assembled using the SeqMan program (DNAStar, Madison, WI, USA), and all new sequences were deposited in GenBank. The sequences were checked against existing sequences at GenBank, and related sequences were obtained for each analysis.

Dataset 1, a combined dataset of nrITS-LSU genes (91 sequences), was composed of 10 newly generated sequences (8 ITS, 2 LSU) and 81 sequences from GenBank (54 ITS, 27 LSU). Totally; 63 sequences in the *Agaricus* clade of the Agaricaceae were included, representing *Agaricus*, *Clarkeinda*, *Coniolepiota*, *Eriocybe*, *Heinemannomyces*, *Hymenagaricus*, *Pseudolepiota*, *Micropsalliota,* and *Xanthagaricus*; and *Chlorophyllum rhacodes* (Vittad.) Vellinga is set as outgroup (Table 1). Dataset 2, *Chlorophyllum* species, 125 sequences of nrITS sequences, was composed of 10 new sequences and 115 sequences from GenBank representing all known species in the genus; *Clarkeinda trachodes* (Berk.) Singer and *Agaricus campestris* L. are outgroups, and GenBank accession number and country of origin are indicated after the species name in the tree (Figure 2). Dataset 3, a combined dataset of nrITS-LSU-*rpb2* genes (139 sequences) of *Chlorophyllum* species, was composed of 19 new sequences (10 nrITS, 8 LSU, 1 *rpb2*) and 120 sequences from GenBank (48 nrITS, 40 LSU, 32 *rpb2*), with *Clarkeinda trachodes* and *Agaricus campestris* as outgroup (Table 2); all known *Chlorophyllum* taxa are represented in this dataset as well. Dataset 4, a dataset of nrITS gene regions (96 sequences) of *Macrolepiota* species, was composed of 17 new sequences

and 80 sequences from GenBank, with *Leucoagaricus meleagris* as the outgroup, and Gen-Bank accession number and country of origin are indicated after the species name in the tree (Figure 4).

**Table 1.** List of collections of taxa in the Agaricaceae and their GenBank accession numbers used in the molecular analyses of dataset 1.

| Table 6242. | Voucher Number | Country/Collection Number | GenBAnk Accession Number | |
| --- | --- | --- | --- | --- |
| | | | ITS | LSU |
| *Agaricus* **aff.** *campestris* | **Murphy 6242** | USA | HM488744 | - |
| *Agaricus bisporatus* | Contu1 | - | AF432882 | - |
| *Agaricus bohusii* | LAPAG562 | - | KM657928 | KR006613 |
| *Agaricus deserticola* | M. Smith | USA | HM488747 | - |
| *Agaricus diminutivus* | Vellinga 2360 | USA | AF482831 | AF482877 |
| *Agaricus megacystidiatus* | MFLU 12-0137 | Thailand | NR_119953 | - |
| *Agaricus* sp. | NTS113 | Thailand | JF514531 | - |
| *Agaricus* sp. | C3182 | Togo | KJ540956 | - |
| *Agaricus* sp. | BAB5059 | India | KR155104 | - |
| *Chlorophyllum rhacodes* | Vellinga 2106 | Netherlands | AF482849 | - |
| *Clarkeinda trachodes* | Iqbal 806 | Bangladesh | - | MG462712 |
| *Clarkeinda trachodes* | ecv3838 | Thailand | HM488750 | HM488771 |
| *Clarkeinda trachodes* | ecv3550 | Thailand | HM488751 | KY418837 |
| *Clarkeinda trachodes* | MFLUCC:100139 | Sri Lanka | KU845621 | - |
| *Clarkeinda trachodes* | **MFLU 19-2351** | **Thailand** | **MN099351** | **-** |
| *Coniolepiota spongodes* | ecv3898 | Thailand | HM488755 | - |
| *Coniolepiota spongodes* | HKAS 77574 | Bangladesh | KC625531 | KC625530 |
| *Eriocybe chionea* | ecv3560 | Thailand | HM488752 | HM488773 |
| *Heinemannomyces splendidissimus* | ecv3586 | Thailand | HM488760 | HM488769 |
| *Heinemannomyces splendidissimus* | zrl3043 | Thailand | JF691559 | - |
| *Heinemannomyces splendidissimus* | GDGM 46633 | China | MF621038 | MF621039 |
| *Hymenagaricus ardosiicolor* | LAPAF9 | Togo | JF727840 | - |
| *Hymenagaricus ardosiicolor* | isolateZ4 | Tanzania | KM360160 | - |
| *Hymenagaricus* cf. *kivuensis* | BR6089 | Burundi | KM982454 | - |
| *Hymenagaricus* sp. | CA833 | Thailand | JF727858 | - |
| *Hymenagaricus* sp. | CA801 | Thailand | JF727859 | |
| *Hymenagaricus* sp. | zrl3103 | Thailand | KM982450 | KM982452 |
| *Hymenagaricus* sp. | LD2012186 | Thailand | KM982451 | KM982453 |
| *Micropsalliota arginophaea* | SFSU zrl 2089 | China | HM436616 | HM436578 |
| *Micropsalliota pusillissima* | SFSU zrl 3047 | China | HM436645 | HM436594 |
| *Pseudolepiota zangmui* | Z. W. Ge 3537 | China | KY768925 | - |
| *Pseudolepiota zangmui* | Z. W. Ge 2106 | China | KY768927 | - |
| *Pseudolepiota zangmui* | Z. W. Ge 2107 | China | KY768926 | - |
| *Pseudolepiota zangmui* | Z. W. Ge 2175 | China | KY768928 | MG742049 |
| *Pseudolepiota zangmui* | **MFLU 10-0515** | **Thailand** | **KX904355** | **-** |
| *Pseudolepiota zangmui* | **MFLU 10-0518** | **Thailand** | **KX904356** | **-** |
| *Pseudolepiota zangmui* | **MFLU 19-2355** | **Thailand** | **MN099352** | **-** |
| *Xanthagaricus caeruleus* | GDGM 50651 | China | MF039088 | MF039086 |
| *Xanthagaricus caeruleus* | GDGM 50794 | China | MF039089 | MF039087 |
| *Xanthagaricus epipastus* | zrl 3045 | Thailand | HM436649 | HM436609 |
| *Xanthagaricus flavosquamosus* | GDGM 50913 | China | MF351627 | - |
| *Xanthagaricus flavosquamosus* | GDGM 50924 | China | MF351628 | - |
| *Xanthagaricus flavosquamosus* | GDGM 50918 | China | MF351629 | MF351631 |
| *Xanthagaricus ianthinus* | HMJAU45193 | China | MH166760 | - |
| *Xanthagaricus ianthinus* | HMJAU45192 | China | MH166761 | - |
| *Xanthagaricus ianthinus* | HMJAU45191 | China | MH166762 | - |
| *Xanthagaricus ianthinus* | HMJAU45194 | China | MH166764 | - |
| *Xanthagaricus necopinatus* | Iqbal821 | Bangladesh | MF351626 | MF351630 |

| *Xanthagaricus necopinatus* | **MFLU 19-2353** | **Thailand** | **MN480544** | **-** |
|---|---|---|---|---|
| *Xanthagaricus necopinatus* | **MFLU 19-2358** | **Thailand** | **MN480545** | **-** |
| *Xanthagaricus pakistanicus* | LAH SH 207 | Pakistan | KY621555 | KY621554 |
| *Xanthagaricus pakistanicus* | HUP SH 315 | Pakistan | KY621556 | - |
| *Xanthagaricus pakistanicus* | SWAT SH 389 | Pakistan | KY621557 | - |
| *Xanthagaricus purpureosquamulosus* | **MFLU 19-2354** | **Thailand** | **MN099353** | **MN097917** |
| *Xanthagaricus purpureosquamulosus* | **MFLU 19-2356** | **Thailand** | **MN099354** | **MN097918** |
| *Xanthagaricus siamensis* | MFLU 19-0575 | Thailand | MN176991 | MN176981 |
| *Xanthagaricus siamensis* | MFLU 19-0574 | Thailand | MN176992 | MN176982 |
| *Xanthagaricus siamensis* | MFLU 19-0576 | Thailand | MN176993 | MN176983 |
| *Xanthagaricus* sp. | ecv3807 | Thailand | HM488761 | HM488770 |
| *Xanthagaricus taiwanensis* | C.M. Chen 3636 | Taiwan | DQ006271 | DQ006270 |
| *Xanthagaricus taiwanensis* | HKAS 42545 | Taiwan | DQ490633 | - |
| *Xanthagaricus thailandensis* | CMUJK010 | Thailand | MG256663 | MG256665 |
| *Xanthagaricus thailandensis* | CMUJK0115 | Thailand | MG256664 | MG256666 |

**Note:** New specimens and sequences are in bold.

**Table 2.** List of fungal taxa of second dataset of *Chlorophyllum* species and their GenBank accession numbers used in molecular analysis of datasets 2.

| Taxon | Voucher/Collection Number | Country | GenBank Accession Number | | |
|---|---|---|---|---|---|
| | | | ITS | LSU | *rpb2* |
| ***Agaricus campestris*** | **LAPAG370** | China | KM657927 | KR006607 | KT951556 |
| *Chlorophyllum africanum* | PREM 62140 | South Africa | MG741961 | MG742041 | MG742070 |
| *Chlorophyllum africanum* | PREM 62141 | South Africa | MG741963 | MG742042 | MG742071 |
| *Chlorophyllum agaricoides* | HKAS 101312 | Russia | MG742003 | MG742020 | MG742050 |
| *Chlorophyllum agaricoides* | HMAS 71678 | China | MG742004 | MG742021 | MG742051 |
| *Chlorophyllum arizonicum* | AH31724 | Mexico | KR233490 | KR233499 | - |
| *Chlorophyllum arizonicum* | Trappe 11481 | USA | HQ020416 | HQ020419 | - |
| *Chlorophyllum brunneum* | HKAS 101315 | Italy | MG742013 | MG742022 | MG742052 |
| *Chlorophyllum brunneum* | ecv2361 | USA | AY083206 | AF482886 | HM488804 |
| *Chlorophyllum demangei* | Z. W. Ge 3112 | China | MG741965 | MG742027 | MG742056 |
| *Chlorophyllum demangei* | Z. W. Ge 3574 | China | MG741964 | MG742025 | MG742055 |
| *Chlorophyllum demangei* | ecv3622 | Thailand | HM488765 | - | - |
| ***Chlorophyllum demangei*** | **MFLU 12-1769** | **Thailand** | **KJ524556** | **MN097907** | **-** |
| ***Chlorophyllum demangei*** | **MFLU 09-0005** | **Thailand** | **KJ524555** | **MN097908** | **-** |
| *Chlorophyllum demangei* | ecv3557 | Thailand | MN582745 | - | - |
| *Chlorophyllum globosum* | Z. W. Ge 2006-1 | China | MG741995 | MG742023 | - |
| *Chlorophyllum globosum* | PREM 62147 | South Africa | MG742002 | MG742024 | MG742053 |
| ***Chlorophyllum globosum*** | **MFLU 12-1815** | **Thailand** | **KJ524553** | **MN097909** | **-** |
| *Chlorophyllum hortense* | HKAS 101317 | China | MG741967 | MG742026 | MG742054 |
| *Chlorophyllum hortense* | Z. W. Ge 3115 | China | MG741968 | MG742028 | MG742057 |
| *Chlorophyllum hortense* | HKAS 90470 | China | MG741971 | MG742029 | MG742058 |
| ***Chlorophyllum hortense*** | **MFLU 12-1783** | **Thailand** | **KJ524554** | **MN097910** | **-** |
| ***Chlorophyllum hortense*** | **MFLU 19-2352** | **Thailand** | **MN099355** | **MN097916** | **MN816433** |
| *Chlorophyllum hortense* | PC17 | Philippines | MN099356 | MN097915 | - |
| *Chlorophyllum lusitanicum* | AH45540 | Spain | KR233482 | KR233491 | - |
| *Chlorophyllum lusitanicum* | AH43927 | Spain | KR233483 | KR233492 | - |
| *Chlorophyllum molybdites* | HKAS 45051 | China | MG741985 | MG742030 | MG742059 |
| *Chlorophyllum molybdites* | Z. W. Ge 3381 | USA | MG741993 | MG742034 | MG742063 |
| *Chlorophyllum molybdites* | Z. W. Ge 3146 | China | MG741987 | MG742031 | MG742060 |
| *Chlorophyllum molybdites* | HKAS 101322 | Italy | MG741988 | MG742032 | MG742061 |
| *Chlorophyllum molybdites* | Z. W. Ge 3377 | USA | MG741992 | MG742033 | MG742062 |
| ***Chlorophyllum molybdites*** | **MFLU 12-1772** | **Thailand** | **KJ524557** | **MN097911** | **-** |
| ***Chlorophyllum molybdites*** | **MFLU 12-1765** | **Thailand** | **KJ524559** | **MN097912** | **-** |
| ***Chlorophyllum molybdites*** | **MFLU 12-1775** | **Thailand** | **KJ524558** | **MN097913** | **-** |
| *Chlorophyllum molybdites* | S57 | Sudan | MK541941 | - | - |

| | | | | | |
|---|---|---|---|---|---|
| *Chlorophyllum molybdites* | J. States AEF1097 | USA | AF482836 | - | - |
| *Chlorophyllum molybdites* | DMSC09538 | Thailand | KP229775 | - | - |
| *Chlorophyllum molybdites* | AM150 (CUH) | India | KM190077 | - | - |
| *Chlorophyllum molybdites* | HKAS 101319 | Thailand | MG741994 | - | - |
| *Chlorophyllum molybdites* | FS10 | India | MK855510 | - | - |
| *Chlorophyllum neomastoideum* | HKAS 83208 | China | MG741976 | MG742035 | MG742064 |
| *Chlorophyllum nothorachodes* | H. Lepp 1142 | Australia | AF482855 | - | - |
| *Chlorophyllum olivieri* | HKAS 31587 | Germany | MG742016 | MG742036 | MG742065 |
| *Chlorophyllum olivieri* | HKAS 53466 | Germany | MG742017 | MG742037 | MG742066 |
| *Chlorophyllum palaeotropicum* | PREM 62142 | South Africa | MG741978 | MG742038 | MG742067 |
| *Chlorophyllum palaeotropicum* | PREM 62145 | South Africa | MG741982 | MG742039 | MG742068 |
| *Chlorophyllum palaeotropicum* | HKAS 93747 | Benin | MG741983 | MG742040 | MG742069 |
| *Chlorophyllum palaeotropicum* | pc21 | Philippines | MN099357 | MN097914 | - |
| *Chlorophyllum pseudoglobosum* | AM155 | India | KP642506 | KR080484 | - |
| *Chlorophyllum rhacodes* | ecv2106 | Netherlands | AF482849 | AY176345 | - |
| *Chlorophyllum rhacodes* | OKM19588 | USA | U85312 | U85277 | HM488803 |
| *Chlorophyllum sphaerosporum* | HMAS 66153 | China | MG742011 | MG742043 | MG742072 |
| *Chlorophyllum sphaerosporum* | HMAS 71683 | China | MG742012 | MG742044 | MG742073 |
| *Chlorophyllum subrhacodes* | Z. W. Ge 3411 | USA | MG741975 | MG742045 | MG742074 |
| *Chlorophyllum subrhacodes* | Z. W. Ge 3232 | USA | MG741973 | MG742046 | MG742075 |
| *Chlorophyllum subrhacodes* | Z. W. Ge 3385 | USA | MG741972 | MG742048 | MG742077 |
| *Chlorophyllum subrhacodes* | Z. W. Ge 3242 | USA | MG741974 | MG742047 | MG742076 |
| *Clarkeinda trachodes* | ecv3550 | Thailand | HM488751 | KY418837 | HM488802 |

**Note:** New specimens and sequences are in bold.

The datasets were aligned using MAFFT version 7.130-win32 (https://wiki.anunna.wur.nl/index.php/MAFFT_7.130, accessed on 10 December 2021) [40,41]. The final alignments were submitted to the TreeBase website (https://www.tree-base.org/treebase-web/home.html, TreeBase No. 24630, accessed on 10 December 2021).

A Maximum likelihood (ML) analysis was performed in RAxML 7.2.6 [42] for the individual dataset with GTRGAMMA + I as the model of evolution, and branch support was estimated over 1000 bootstrap partitions (BP) with the rapid bootstrap option. For Bayesian inferences (BI) analysis, the best substitution model of the individual dataset was defined by using MrModelTest v.2.3 [43]. For dataset (1), the best-selected model was GTR + I + G for both ITS and LSU. For dataset (2), the ITS dataset, the best-selected model was HKY + I + G. For dataset (3), the best-selected model for ITS was HKY + I + G, and GTR + I + G was the best for LSU and *rpb2*. For dataset (4), the best-selected model was again GTR + I + G. A Bayesian inference (BI) analysis was performed with mrbayes 3.2.7a [44] at the CIPRES webserver (available at https:// www.phylo.org/ (accessed on 10 December 2021) with the suitable setting of 4 individual runs. The Bayesian inference analysis; each one beginning from random trees with 4 simultaneous independent chains, each of 4 Markov chain Monte Carlo (MCMC) chains for 2 independent runs, 2,000,000 generations for datasets 1, 2, and 3, and 1,000,000 generations for dataset (4), sampled every 1000 generations, the first 25% (0.25) of the sampled trees were discarded as burn-in, and all sampled after the average standard deviation of split frequencies lowered than 0.01 were used to reconstruct a 50% majority-rule consensus tree and to calculate the Bayesian posterior probabilities (BPP) of the clades. The phylogenetic tree results of all analyses were exported and edited in TreeView 1.0.0.0. [45]. The phylogenetic trees were edited in the software of Adobe Illustrator CS3.

## 3. Results

### 3.1. Result of the Phylogenetic Analyses

Figure 1 shows the maximum likelihood phylogenetic tree of the extended *Agaricus* clade based on a combined ITS and LSU data set. A total of 63 taxa were analyzed (Table 1). The alignment dataset comprised 1560 characters (including the gaps). Seven different

major clades can be recognized in the phylogenetic tree (Figure 1). The *Xanthagaricus* clade received high bootstrap support and was completely comprised of *Xanthagaricus* species; the two Thai specimens of *X. purpureosquamulosus* had identical sequences and are basal to a clade comprised of *X. epipastus*, *X. necopinatus*, and *X. pakistanicus*; the sequences of the Thai specimens of *Ps. Zangmui* were identical with those from China. *Clarkeinda* also forms a monophyletic clade, composed of a Thai specimen from this study, two Thai specimens from Vellinga et al. [2], a specimen from Bangladesh Hosen et al. [34], and a specimen from the type location in Sri Lanka [46].

Figure 2 indicates the maximum likelihood phylogenetic tree of *Chlorophyllum* species based on nrITS sequences. Data from 126 collections were included, and the alignment dataset comprised 809 characters (including the gaps). Six clades were distinguished, all receiving high bootstrap support. Thai taxa clustered into two sections as in an analysis of combined sequences data (Figure 3).

Figure 3 shows the maximum likelihood phylogenetic tree of *Chlorophyllum* species based on multi-gene DNA data set of ITS, LSU, and *rpb2* gene regions. The alignment comprised 58 collections (Table 2), and this dataset comprised 2322 characters (including the gaps). The six clades or sections in *Chlorophyllum*, (*Chlorophyllum*, *Ellipsoidospororum*, *Endoptychorum*, *Rhacodium*, *Parvispororum*, and *Sphaerospororum*) received high bootstrap support, and these results are consistent with those found by Ge et al. [10]. The Thai specimens were distributed in two sections: *C. demangei* and *C. hortense* with white basidiospores without a germ pore in section *Ellipsoidospororum*, and the species with greenish basidiospores and a germ pore, *C. globosum* and *C. molybdites,* in section *Chlorophyllum*.

Figure 4 shows the maximum likelihood phylogenetic tree of *Macrolepiota* species based on nrITS sequences. A total of 97 sequences was analyzed; the aligned dataset comprised 777 characters (including the gaps). Three major clades or sections (sect. *Macrolepiota*, sect. *Macrosporae*, sect. *Volvata*) were recognized with high bootstrap support in the phylogenetic tree.

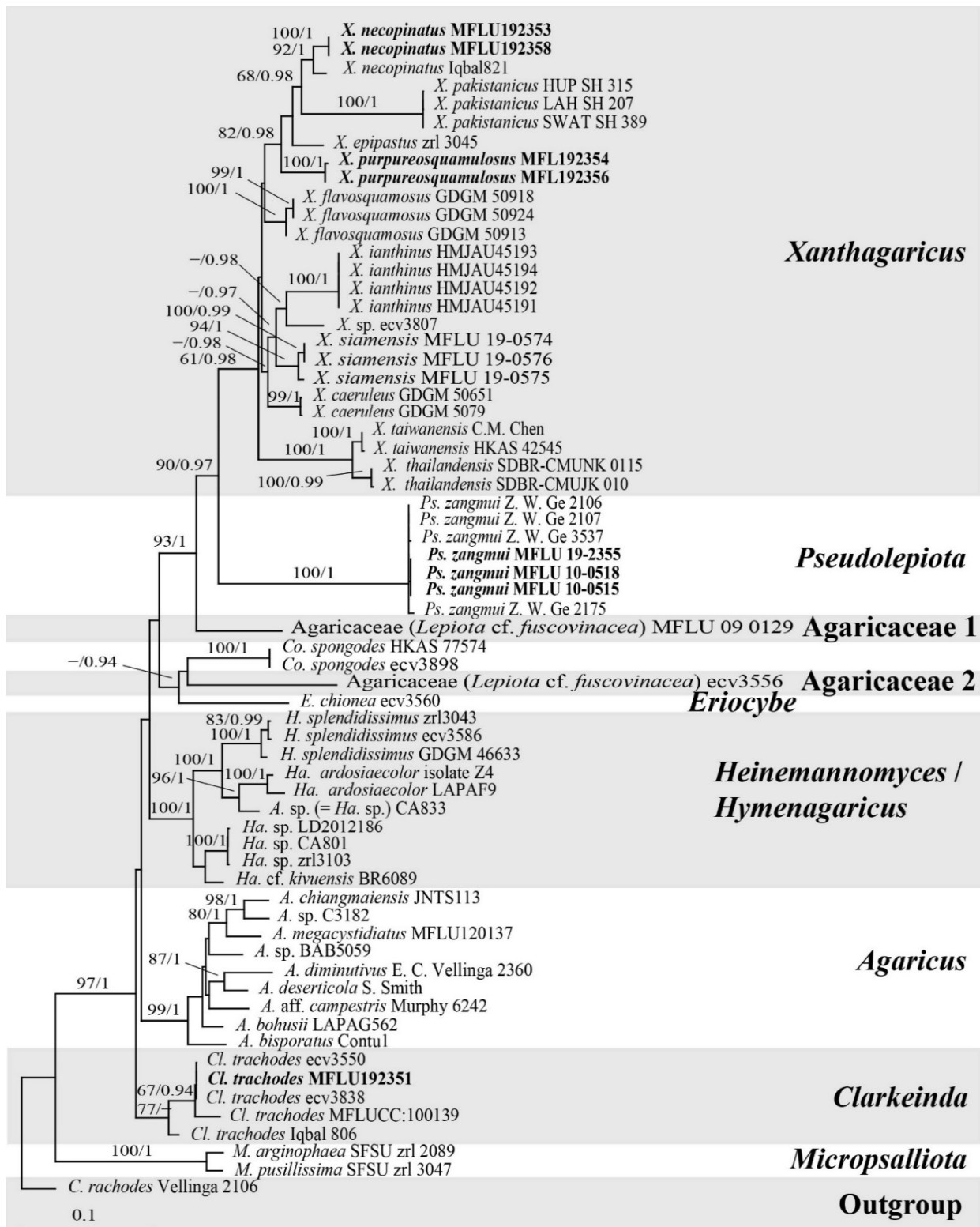

**Figure 1.** Maximum likelihood phylogenetic tree of the *Agaricus* clade in the Agaricaceae based on nrITS-LSU sequence data. New sequences generated from Thailand are in bold. Bootstrap values (ML/B) are given above the branches. Bootstrap values for maximum likelihood equal to or greater than ≥60 and Bayesian posterior probabilities ≥ 0.95 are placed above the branches. Genus abbreviations are as follows: *A* = *Agaricus*, *C* = *Chlorophyllum*, *Cl* = *Clarkeinda*, *Co* = *Coniolepiota*, *E* = *Eriocybe*, *H* = *Heinemannomyces*, *Ha* = *Hymenagaricus*, *Ps* = *Pseudolepiota*, *M* = *Micropsalliota*, and *X* = *Xanthagaricus*. *Chlorophyllum rachodes* Vellinga is an outgroup.

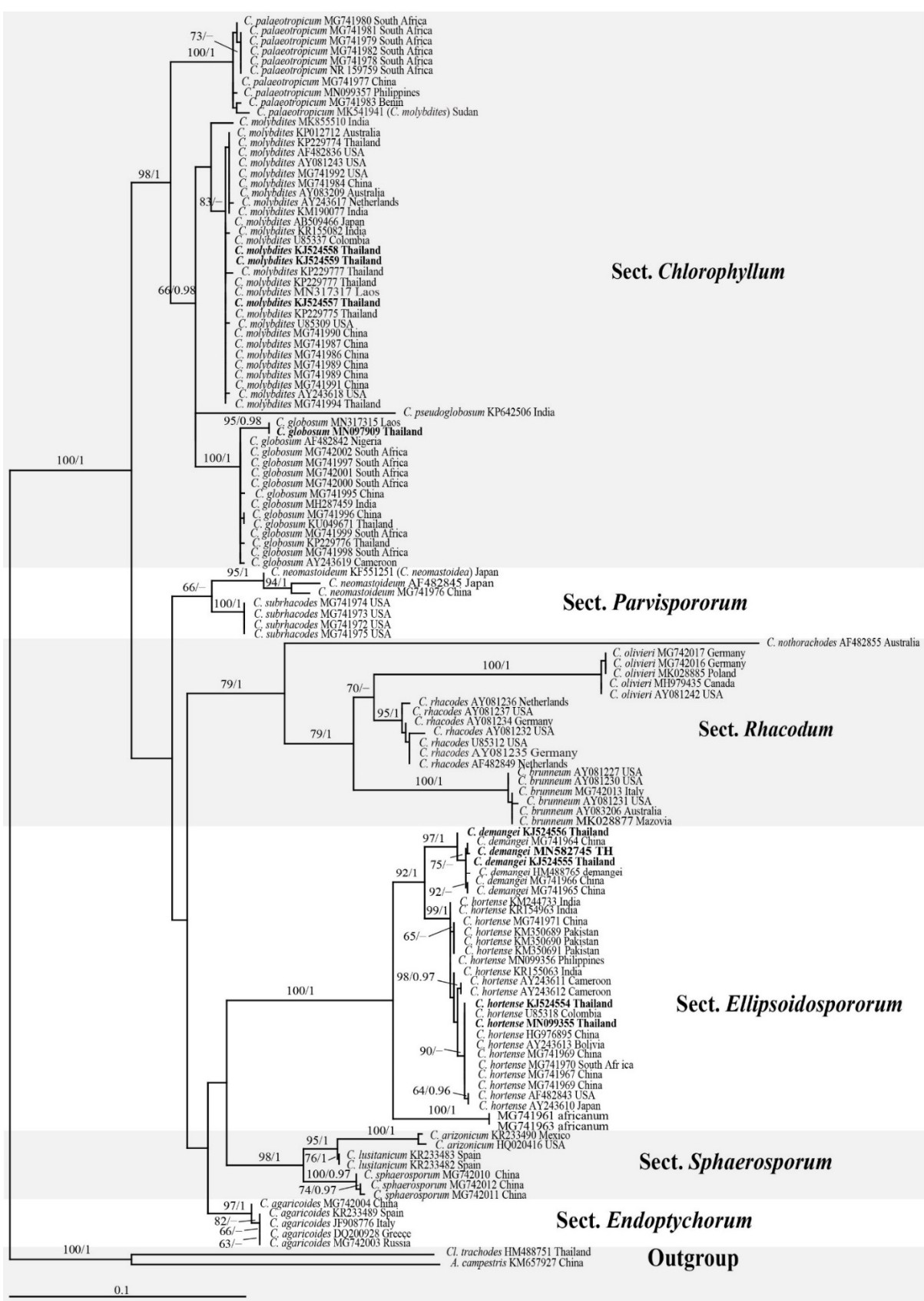

**Figure 2.** Maximum likelihood phylogenetic tree of *Chlorophyllum* based on nrITS sequences. New sequences generated from Thailand are in bold. Bootstrap values for maximum likelihood equal to or greater than ≥ 60 and Bayesian posterior probabilities ≥ 0.95 are placed above the branches. Genus abbreviations are as follows: *A= Agaricus, C= Chlorophyllum, Cl= Clarkeinda*, Thai specimens from this study are in bold types. *Clarkeinda trachodes* (Berk.) Singer and *Agaricus campestris* L. are outgroup. GenBank accession number and country are indicated after species name.

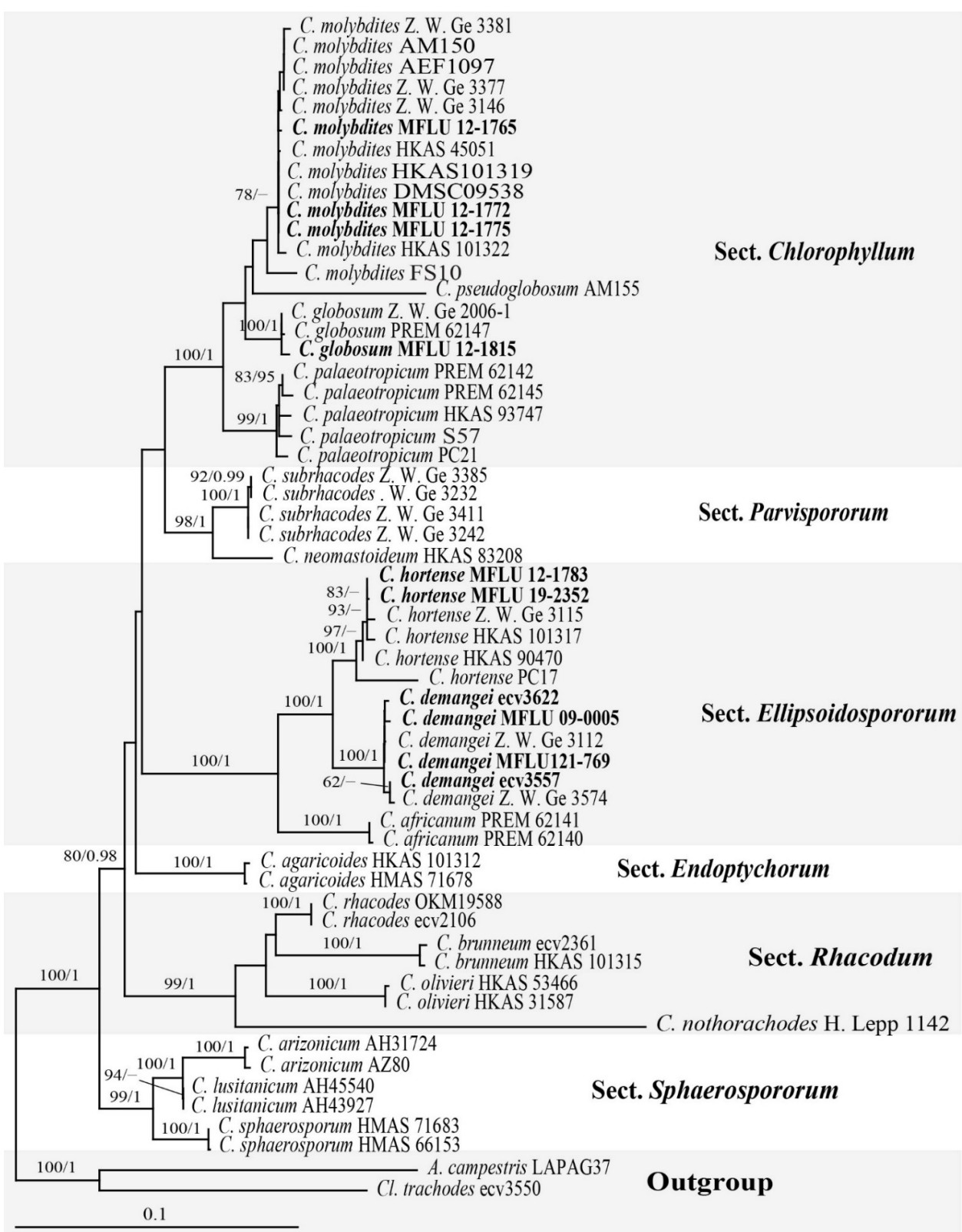

**Figure 3.** Maximum Likelihood tree of *Chlorophyllum* based on analysis of nrITS-LSU-*rpb2* sequence data. Bootstrap values (ML/B) are given above the branches. Bootstrap values for maximum likelihood equal to or greater than ≥60 and Bayesian posterior probabilities ≥ 0.95 are placed above the branches. Abbreviations are as follows: *A.* = *Agaricus, C* = *Chlorophyllum,* and *Cl* = *Clarkeinda*. *Clarkeinda trachodes* and *Agaricus campestris* are used as outgroup. New sequences are in bold type.

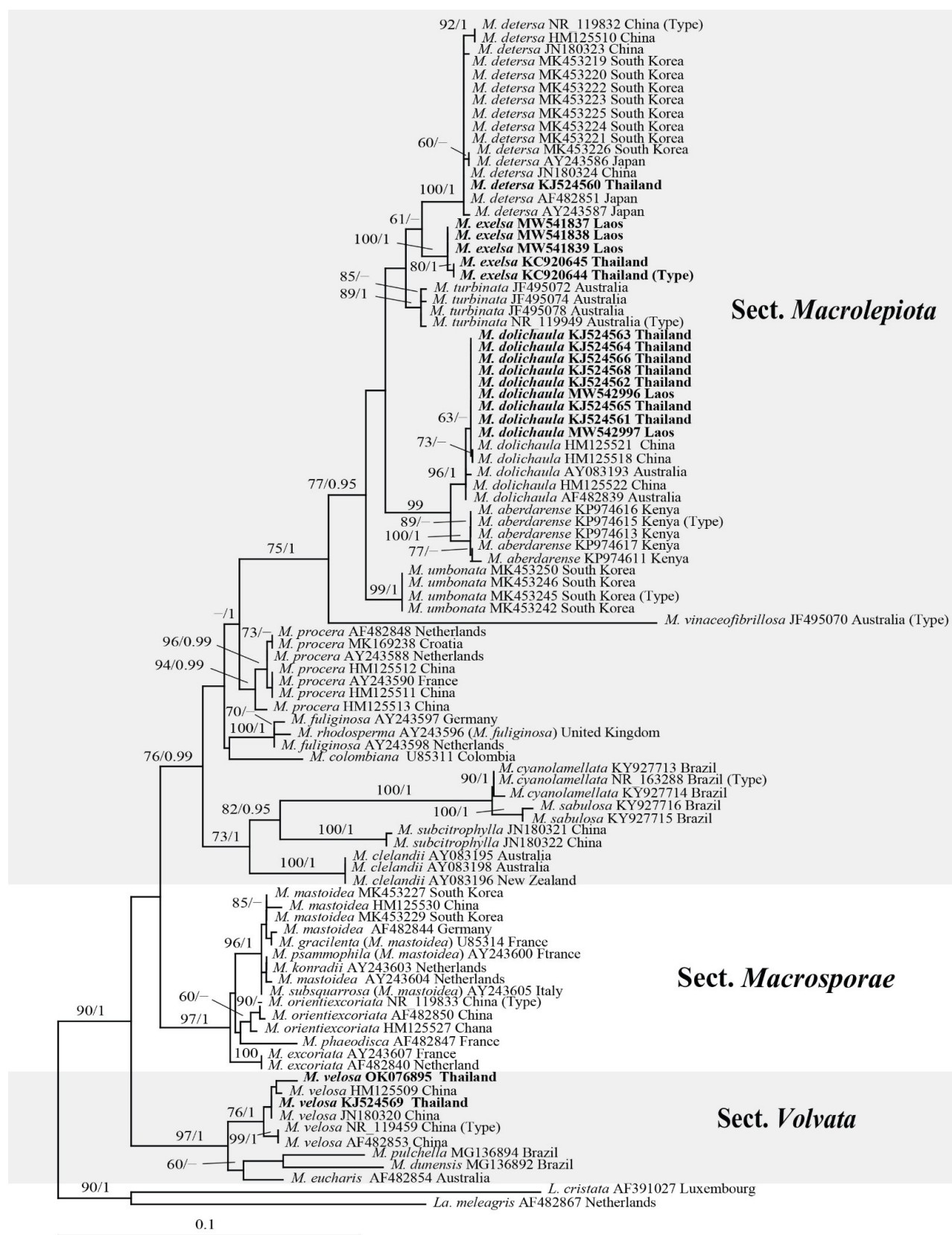

**Figure 4.** Maximum parsimony tree of *Macrolepiota* based on analysis of nrITS sequences. Bootstrap values for maximum likelihood equal to or greater than ≥ 60 and Bayesian posterior probabilities ≥ 0.95 are placed above the branches. Abbreviations are as follows: *La.* = *Leucoagaricus*, *L* = *Lepiota*, and *M* = *Macrolepiota*. All taxon names in bold are new sequences in this study. *Leucoagaricus meleagris* (Gray) Singer and *Lepiota cristata* (Bolton) P. Kumm. are used as outgroup. Country, type, and GenBank accession number are indicated after each species name.

*3.2. Taxonomy*

3.2.1. Chlorophyllum Massee

*Chlorophyllum demangei* (Pat.) Z.W. Ge & Zhu L. Yang, MycoKeys 33: 80 (2018)

    ≡ *Lepiota demangei* Pat., Bull. trimest. Soc. mycol. Fr. 23 (2): 78 (1907).

    Index Fungorum number: IF 823863; Facesoffungi number: FoF 07067; Figures 5 and 6.

Pileus 50–90 mm, subglobose when young, expanding to parabolic, convex-hemispherical with umbo, applanate to plano-concave, with straight to the deflexed margin; surface smooth when young, brownish-grey (5D2), later breaking up into scales and leaving light-brown to brown (5D4, 6D7–8) glabrous calotte at the center, with concolorous with tufted patches around umbo and irregular patches scattered toward the margin, on white to pale-yellow (5A3) background, turning orange-white (5A2) with time; margin with strongly sulcate marginal zone, striate, up to 15 mm wide, with exceeding lamellae when mature. Lamellae free, slightly or distinctly ventricose, up to 5 mm wide, white to yellowish-white (4A2), crowded, lamella-edge eroded. Stipe 10–90 × 6–7 mm, cylindrical, or slightly wider at base; surface covered with white fibrils, turning orange-white (5A2) when touched. Annulus superonate, attached at the upper side of the stipe, moveable when mature, with surface concolorous to patches on the pileus. The context in pileus white, in stipe white and hollow, turning orange-white (5A2) in both stipe and pileus context. Taste peanut-like. Smell mild. Spore print white.

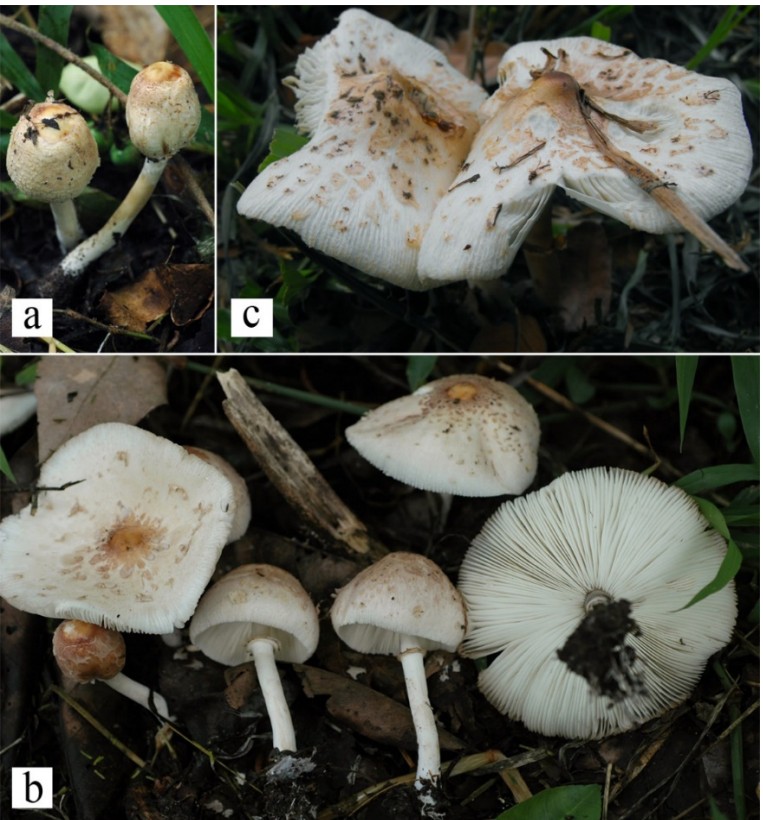

**Figure 5.** Fresh basidiomata of *Chlorophyllum demangei* in situ. (**a**,**b**) MFLU 09-0005. (**c**) MFLU 12-1769.

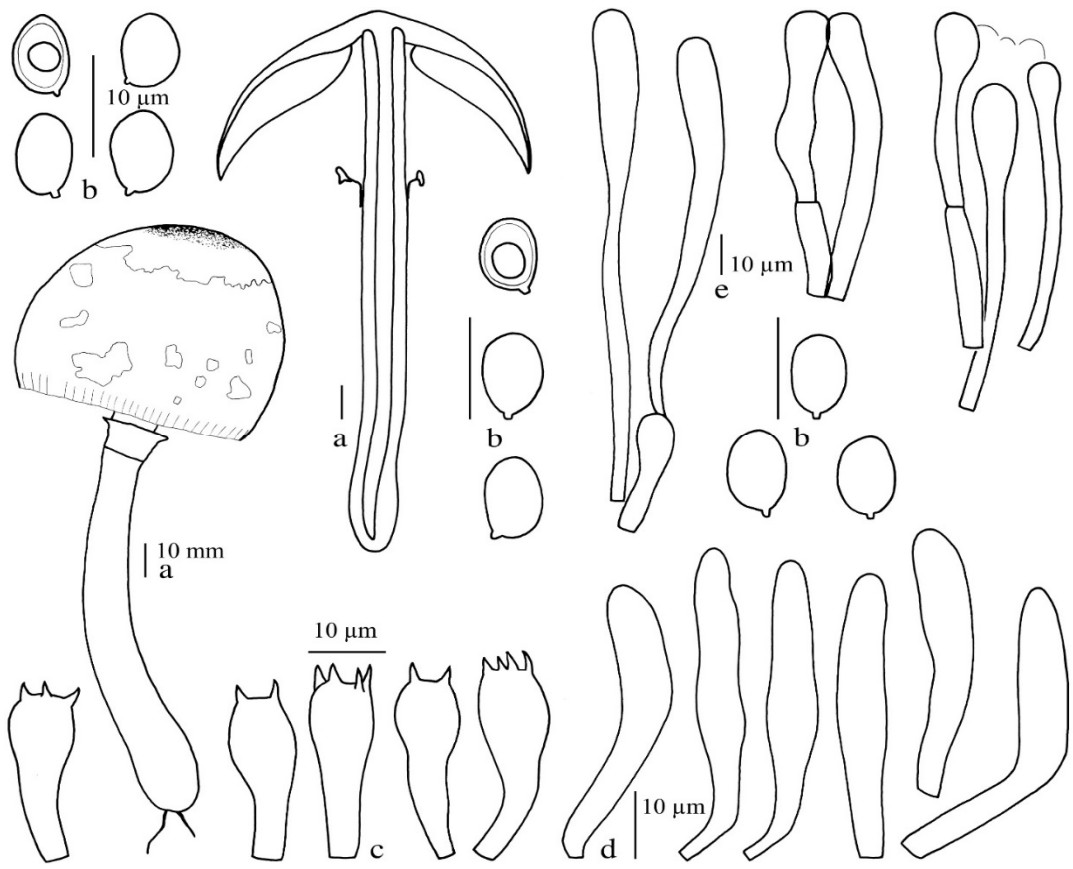

**Figure 6.** *Chlorophyllum demangei* (ecv3622). (**a**) Basidiomata. (**b**) Basidiospores. (**c**) Basidia. (**d**) Cheilocystidia. (**e**) Pileus covering elements.

Basidiospores [100,4,4] 6.5–8.5 × 4.2–6.5 μm, avl × avw = 7.5 × 5.5 μm, Q = 1.3–1.67, Qav = 1.4–1.55, in side-view ellipsoid to oblong-amygdaliform, in frontal view ellipsoid, oblong, without germ pore, slightly thick-walled, hyaline, dextrinoid, congophilous, cyanophilous, metachromatic in Cresyl Blue. Basidia 18–28 × 7.0–11.5 μm, clavate, hyaline and thin-walled, 2-spored and 4-spored, occasionally 1-spored. Lamella edge sterile, with abundant cheilocystidia. Cheilocystidia 24–55 × 6–9.5 μm, narrowly clavate, slightly narrowly lageniform, cylindrical, colorless, and slightly thick-walled. Pleurocystidia absent. Pileus covering a trichoderm made up of cylindrical, narrowly clavate elements with a long stalk, 40–145 × 6–14 μm, colorless or with pale-brown parietal pigment, with the encrusted wall in some elements and lower hyphae. Stipe covering a cutis made up of cylindrical hyphae and elements, colorless, up to 12 μm wide. Clamp connections not found.

Habitat and distribution: Solitary or in large groups, saprotrophic, on decaying wood and leaves on rich humus soil; in deciduous rainforest, gardens, and grasslands.

Material examined: Thailand, Chiang Mai Province, Mae Taeng District, Pha Deng Village: 19°07′13.7″ N; 98°43′52.9″ E, alt., 905 m, 6 July 2007, P. Sysouphanthong, PNG05 (MFLU 09-0005); *ibidem*, 12 July 2007, P. Sysouphanthong, PNG050 (MFLU 09-0051); *ibidem*, 25 July 2007, E. C. Vellinga, ecv3557; *ibidem*, 15 July 2007, E. C. Vellinga, ecv3622; Chiang Rai, Muang District, Pong Phra Bath Village, 21 July 2012, P. Sysouphanthong, 2012-6 (MFLU 12-1769).

Notes: the Thai collections of *Chlorophyllum demangei* can be recognized by brownish-grey to yellowish-brown irregular patches on white pileus with sulcate or striate margin in most collections, white and free lamellae with a white spore print, a simple annulus, oblong-ovoid to oblong amygdaliform, and hyaline basidiospores without a germ pore, clavate, 4-spored basidia, narrowly clavate to cylindrical cheilocystidia, trichodermal pileus covering made up of cylindrical to narrowly clavate elements, and absence of clamp connections. It closely resembles *C. hortense*, microscopically because of the basidiospores

without a germ pore and the cylindrical cheilocystidia, but it differs in the 4-spored basidia. Thus far, it seems to be much more restricted in its distribution than *C. hortense*, as it is only known from China, northern Thailand, and Vietnam [10,47]. This is the first record of the species from Thailand.

*Chlorophyllum globosum* (Mossebo) Vellinga, Mycotaxon 83: 416 (2002)

≡ *Macrolepiota globosa* Mossebo, Mycotaxon 76: 268 (2000).
Index Fungorum number: IF 374394; Facesoffungi number: FoF 07069; Figures 7 and 8.

Pileus 50–200 mm, subglobose to globose when young, expanding to parabolic to convex, plano-concave when fully mature, with straight or slightly inflexed margin; glabrous at the center, pale-orange to brownish-orange (5A2–3, 6C7–8), with concolorous patches of irregular shapes around the center and scatted towards the margin and fragile when mature, on white to yellowish-white (2A2, 3A2, 4A2) felted or fibrillose background and turning pastel red to red (9A4–6) when touched; margin white, sulcate, but non-striate, not fringed, exceeding lamellae when mature. Lamellae free, slightly remote from the stipe, white to orange-white (5A1–2) when young, turning pastel red to red (9A4–6) when touched, pastel green to grayish-green (29A4, 29B4) when fully mature, crowded, ventricose, and narrowly to pileal margin, up to 20 mm wide, with the eroded edge. Stipe 85–210 × 12–31 mm, tapering to apex, with bulb-like, 30–34 mm wide base; white to brownish-orange (6C3-6) background, sometimes with some white fibrils at apical zone, turning pastel red to red (9A4–6) when touched, with white rhizomorphs connected to the substrate. Annulus under-developed, cuff-like, moveable when mature, white and fibrillose in the upper part, at underside with squamules similar to the patches on the pileus. Context thick and white in pileus, white in the stipe, brownish-orange (6C3-6) at apical zone, paler in the middle zone, and white toward the base, turning pastel red to red (9A4–6) in both pileus and stipe context. Smell and taste not observed. Spore print dull green to grayish-green (29D3–5, 29D5–6).

Basidiospores [75,3,2] 9.0–12.0 × 7.0–9.0 μm, avl x avw = 10.3–11.3 × 7.4–8.3 μm, Q = 1.3–1.57, avQ = 1.4, in side-view broadly ellipsoid to ellipsoid-amygdaliform, in frontal view broadly ellipsoid or amygdaliform, with truncate apex and germ pore, thick-walled, hyaline to hyaline green, dextrinoid, congophilous, cyanophilous, metachromatic. Basidia 22.0–34.0 × 9.0–14.0 μm, clavate, hyaline, thin-walled, 4-spored, often 2-spored, rarely 1-spored. Lamella edge sterile. Cheilocystidia 30.0–58.0 × 10.0–29.0 μm, essentially clavate, occasionally with a slightly long stalk, rarely utriform and spheropedunculate, thin-walled and colorless. Pleurocystidia absent. Pileus covering a hymenoderm made up of layers of oblong to cylindrical, narrowly clavate elements, terminal elements 40.0–100 × 5.0–11.0 μm, with rounded or attenuate apex, hyaline and thin-walled, sometimes with pale-yellow intracellular pigment in upper elements. Clamp connections not found.

Habitat and distribution: growing in small groups, saprotrophic, on nutrient-rich soil, in grassland in shade of *Samanea saman* and on soil rich in humus in deciduous rain forest during June to August in the wet season of northern Thailand and also known from Vientiane Capital, Laos.

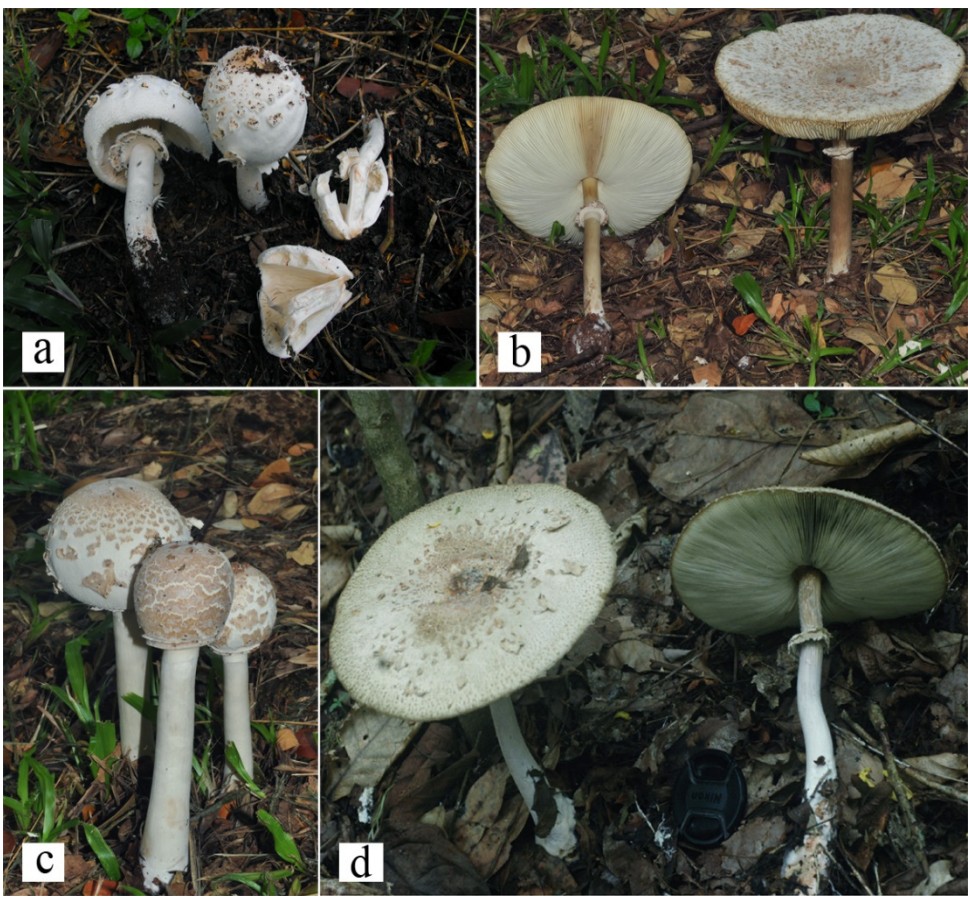

**Figure 7.** Fresh basidiomata of *Chlorophyllum globosum* in situ. (**a**) MFLU 12-1815. (**b**,**c**) MFLU 192357. (**d**) MFLU 10-0555.

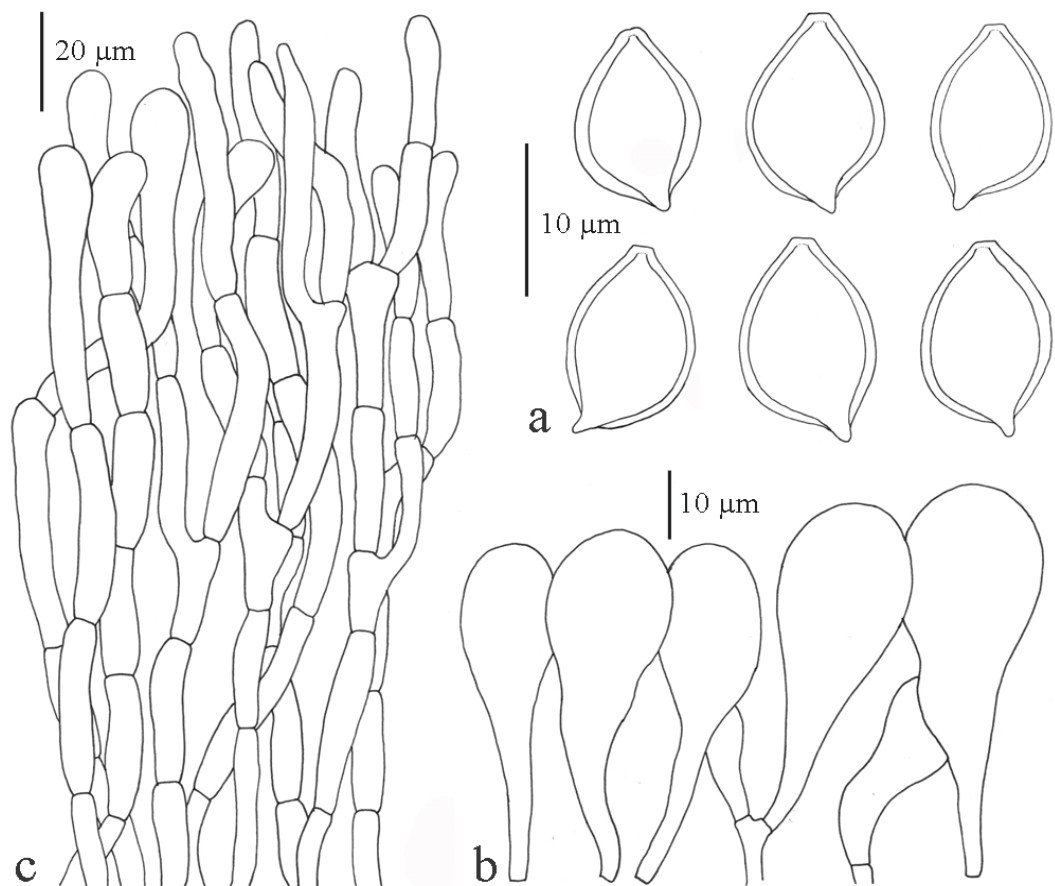

**Figure 8.** *Chlorophyllum globosum* (MFLU 12-1815). (**a**) Basidiomata. (**b**) Basidiospores. (**c**) Pileus covering elements.

Material examined: THAILAND, Chiang Mai Province, Mae Taeng District, Pongduad Village:16°06′16.1″ N; 99°43′07.9″ E, alt., 780–805 m, 16 June 2010, P. Sysouphanthong, P37 (MFLU 10-0555); Chiang Rai Province, Muang District, Ratjabhat University Campus, 30 August 2012, P. Sysouphanthong, 2012-21 (MFLU 12-1815); Chiang Rai Province, Muang District, Forest of Mae Fah Luang University Campus, 10 July 2019, P. Sysouphanthong, 2019-24 (MFLU 19-2357). LAOS, Vientiane Capital, Xaythany, Houay Yang Preserve Forest, 16 June 2017, P. Sysouphanthong, PS2017–7 (HNL503445).

Notes: *Chlorophyllum globosum* is recognized by medium to large-sized, white basidiomata, subglobose to globose pileus, which expands to convex when mature, pale-brown to orange-brown patches on white pileus, a white stipe, a cuff-like annulus, and a reddening reaction in most parts of the basidioma when touched or damaged.

*Chlorophyllum globosum* was described from Cameroon as *Macrolepiota globosa* [48], and the Thai material fits the description very well, except for the spore color, which was originally given as white. The species was transferred to *Chlorophyllum* by Vellinga [49], based on phylogenetic analyses of ITS and LSU. *C. globosum* is very close to *C. molybdites* in morphology, and they are sister taxa in the phylogenetic trees, but they are definitely different species [10,27].

The noticeable differences between the *Chlorophyllum molybdites* and *C. globosum* samples from Thailand and Laos are the whiter basidiomata, longer elements of the pileus covering, and longer cheilocystidia with a distinct stalk in *C. globosum* (Figures 7 and 8), while cheilocystidia are subglobose, broadly clavate to clavate without a long stalk in *C. molybdites* (Figures 11 and 12). However, they are identical in most other characteristics. If we had not obtained sequence data from our collections, this species would most probably not have been recognized, and the different morphology of the cheilocystidia tabled as an intraspecific variation.

*Chlorophyllum globosum* is now known from various countries in Africa (Benin, Cameroon, and South Africa) and Asia (India, Laos, Thailand, and China). The occurrence in some countries is based on ITS sequences in GenBank that are identical to those from Thailand (Figure 2).

*Chlorophyllum hortense* (Murrill) Vellinga, Mycotaxon 83: 416 (2002)

≡ *Lepiota hortensis* Murrill, N. Amer. Fl. 10 (1): 59 (1914)
≡ *Leucoagaricus hortensis* (Murrill) Pegler, Kew Bull. add. Ser. 9: 414 (1983)
≡ *Lepiota humei* Murrill, Lloydia 6: 220 (1943)
= *Chlorophyllum humei* (Murrill) Vellinga, Mycotaxon 83: 416 (2002)
= *Lepiota mammillata* Murrill, Lloydia 6: 220 (1943)
= *Chlorophyllum mammillatum* (Murrill) Vellinga, Mycotaxon 83: 416 (2002)
= *Lepiota subfulvidisca* Murrill, Lloydia 6: 221 (1943)
= *Chlorophyllum subfulvidiscum* (Murrill) Vellinga, Mycotaxon 83: 416 (2002)
= *Lepiota alborubescens* Hongo, Mem. Fac. lib. Arts Educ. Shiga Univ., nat. Sci. 12: 40
= *Macrolepiota alborubescens* (Hongo) Hongo, Trans. Soc. mycol. Japan 27: 107 (1986)
= *Chlorophyllum alborubescens* (Hongo) Vellinga, Mycotaxon 83: 416 (2002)
= *Leucoagaricus bisporus* Heinem., Bull. Jard. bot. nat. Belg. 43: 8 (1973).
Index Fungorum number: IF 374396; Facesoffungi number: FoF 03444; Figures 9 and
10.

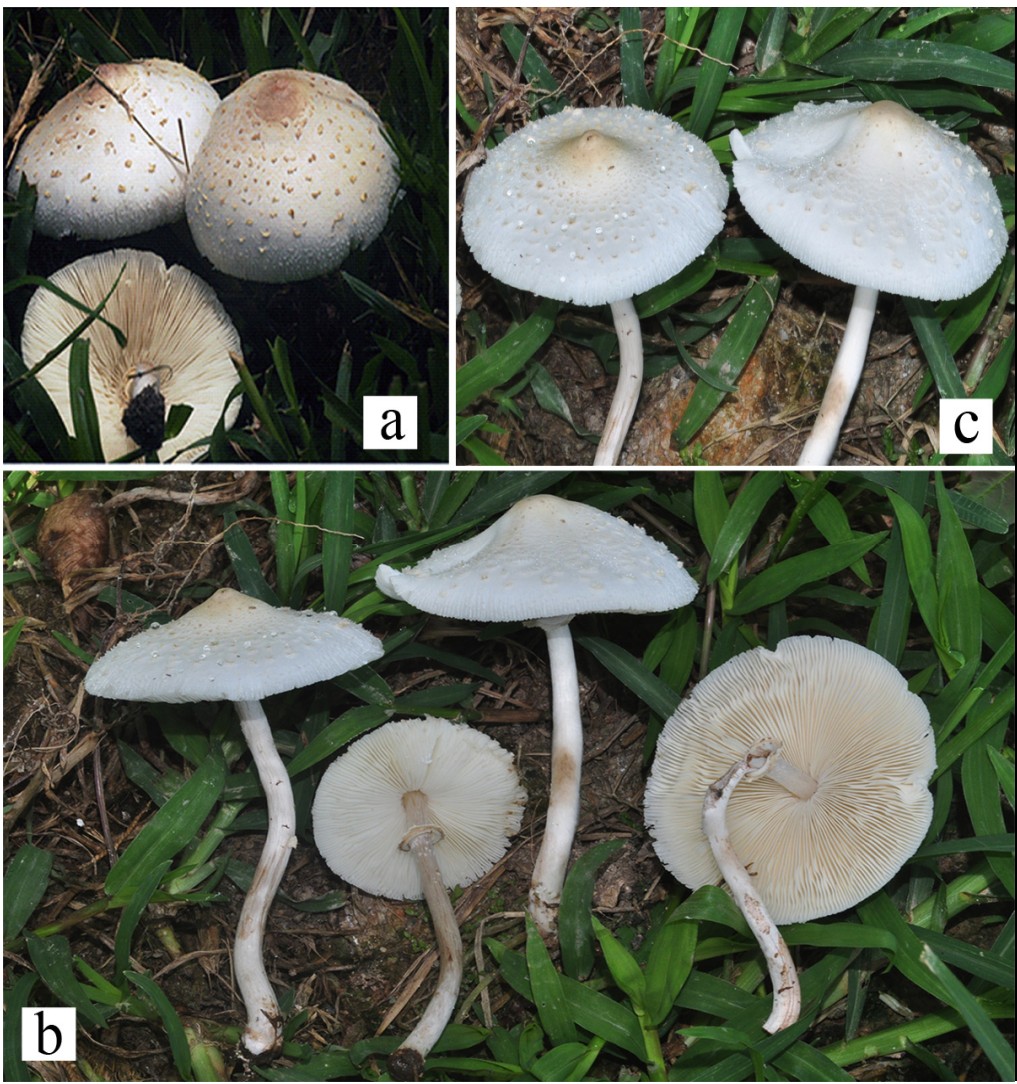

**Figure 9.** Fresh basidiomata of *Chlorophyllum hortense*. (**a**) MFLU 12-1783. (**b**,**c**) MFLU 19-2352.

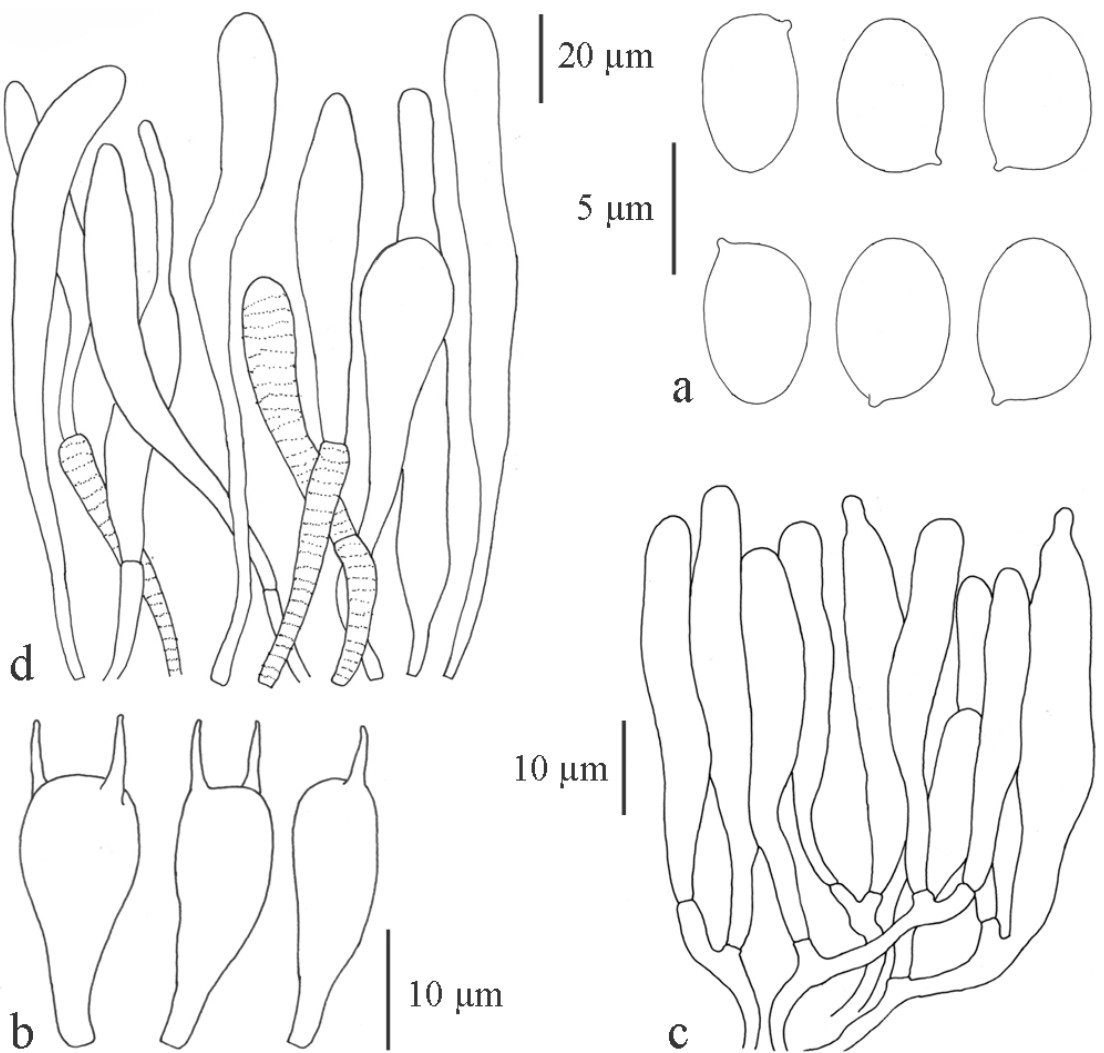

**Figure 10.** Microcharacters of *Chlorophyllum hortense* (MFLU 12-1783). (**a**) basidiospores. (**b**) basidia. (**c**) cheilocystidia. (**d**) pileus covering.

Pileus 72–98 mm, convex, expanding to umbonate with distinctly umbo, with straight margin; surface covered with light-brown to yellow-brown (5D4–5) glabrous calotte at center, with brownish-yellow (5C7–8) irregular patches or squamules toward the margin, on orang white to light-orange (5A2–4) background; margin sulcate or slightly striate, white. Lamellae free, narrowly fusiform, up to 7 mm wide, white, crowded, lamella-edge eroded. Stipe 60–110 × 6–7 mm, cylindrical, slightly wider to base; surface smooth, white. Annulus superonate, moveable, white, with brownish-yellow (5C7–8) on the upper part. Context white in pileus, white in stipe and hollow. All parts of basidiomata turn orange-white (5A2) when touched. Taste peanut-like. Smell mild. Spore print white.

Basidiospores [50,2,1] 6.0–8.5 × 5.3–6.5 μm, avl × avw = 7.5 × 5.8 μm, Q = 1.2–1.54, Qav = 1.3, in side-view broadly ellipsoid to ellipsoid amygdaliform, in frontal view ellipsoid, oblong, without germ pore, hyaline, dextrinoid, congophilous, cyanophilous, metachromatic. Basidia 25–30 × 8–11.5 μm, clavate, hyaline, 2-spored, occasionally 1-spored. Lamella edge sterile, with abundant cheilocystidia. Cheilocystidia 35–50 × 7–9 μm, narrowly clavate, cylindrical, sometimes with short apical excrescence or appendage, colorless. Pleurocystidia absent. Pileus covering of scales a trichoderm made up of cylindrical, narrowly clavate elements with long stalk, 30–155 × 6.3–16 μm, colorless or with pale-brown parietal pigment, with an encrusted wall in some elements and lower hyphae. Stipe covering a cutis made up of cylindrical hyphae and elements, colorless, 10 μm wide. Clamp connections not observed.

Habitat and distribution: growing in large groups, in grasslands, saprotrophic on nutrient-rich soil.

Material examined: Thailand, Chiang Rai Province, Muang District, Mae Korn District, 22 July 2010, P. Sysouphanthong, BJP54 (MFLU 12-1783); Muang District, Pongphabath Village, 1 July 2018, P. Sysouphanthong, PS2018-31 (MFLU 19-2352).

Notes: *Chlorophyllum hortense* can be recognized by light-brown to yellow-brown irregular patches on white to light-orange pileus and sulcate or striate margin, white and free lamellae with a white spore print, an annulus, oblong ovoid to oblong amygdaliform, and hyaline basidiospores without a germ pore, 2-spored basidia, narrowly clavate to cylindrical cheilocystidia, trichodermal pileus covering made up of cylindrical to narrowly clavate elements, and absence of clamp connections. It should be noted that most species of lepiotaceous fungi have 4-spored basidia; only a handful of species in various genera are characterized by 2-spored basidia.

This species has been described by several mycologists [27,50], and because of the basidiospores without a germ pore, it was accommodated in *Leucoagaricus*. It is one of the most widespread species in *Chlorophyllum*. A record from India [51] might refer to *C. demangei*, because of the 4-spored basidia.

The species is found throughout the tropics in diverse, mostly man-influenced habitats, on dung, compost, and soil [14,27].

*Chlorophyllum molybdites* (G. Mey.) Massee, Kew Bull. 1898: 136 (1898)

≡ *Agaricus molybdites* G. Mey., Pr. Fl. essequ.: 300 (1818)

≡ *Lepiota molybdites* (G. Mey.) Sacc., Syll. Fung. 5: 30 (1887)

= *Mastocephalus molybdites* (G. Mey.) Kuntze, Rev. Gen. Pl. 2: 860 (1891)

= *Leucocoprinus molybdites* (G. Mey.) Pat., Bull. Soc. mycol. Fr. 29: 215 (1913)

= *Macrolepiota molybdites* (G. Mey.) Moreno, Bañares and Heykoop, Mycotaxon 55: 467 (1995)

= *Agaricus morganii* Peck, Bot. Gaz. 4: 137 (1879)

= *Lepiota morganii* (Peck) Sacc., Syll. Fung. 5: 30 (1887)

= *Mastocephalus morganii* (Peck) Kuntze, Rev. Gen. Pl. 2: 860 (1891)

= *Chlorophyllum morganii* (Peck) Massee, Kew Bull. 1898: 136 (1898)

= *Agaricus glaziovii* Berk., Vidensk., Meddel. 1879-1880: 32 (1880)

= *Pholiota glaziovii* (Berk.) Sacc., Syll. Fung. 5: 751 (1887)

= *Lepiota ochrospora* Cooke and Massee, Grevillea 21: 73 (1893)

= *Chlorophyllum esculentum* Massee, Kew Bull. 1898: 136 (1898)

= *Lepiota esculenta* (Massee) Sacc. and P. Syd., Syll. Fung. 16: 2 (1902)

= *Agaricus guadelupensis* Pat., Bull. Soc. mycol. Fr. 15: 197 (1899)

= *Annularia camporum* Speg. in An. Mus. nac. Buenos Aires, Ser. II, 6: 117 (1899)[1898]

= *Lepiota camporum* (Speg.) Speg.,Bol. Acad. nac. Cienc. Cordoba 29: 114 (1926)

= *Agaricus congolensis* Beeli in Bull. Soc. roy. bot. Belg. 61: 92 (1928)

= *Chlorophyllum molybdites* var. *congolense* (Beeli) Heinem., Fl. Icon. Champ. Congo 16: 323 (1967).

Index Fungorum number: IF 604726; Facesoffungi number: FoF 03445; Figures 11 and 12.

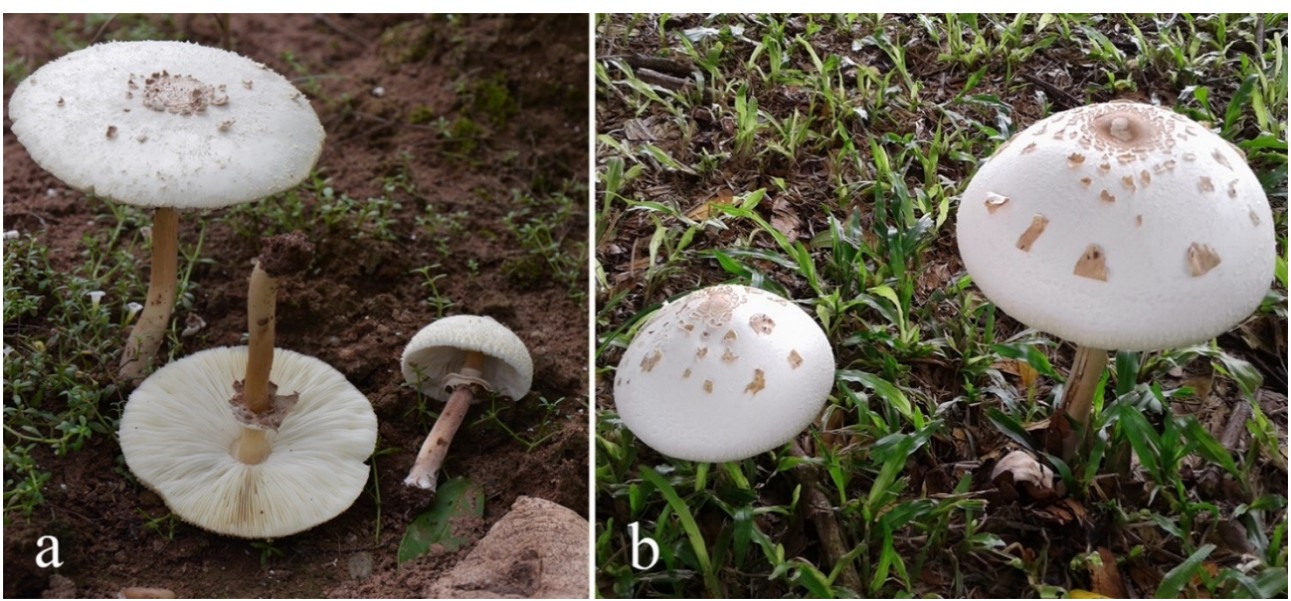

**Figure 11.** Fresh basidiomata of *Chlorophyllum molybdites* in situ. (**a**) MFLU 12-1775. (**b**) MFLU 12-1819.

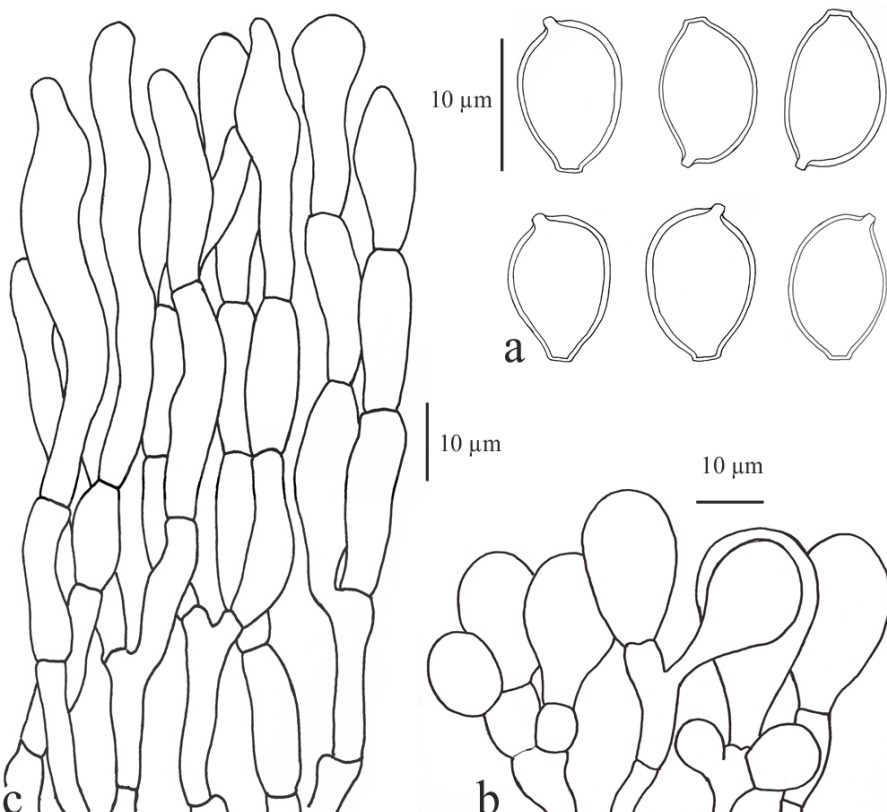

**Figure 12.** Microcharacters of *Chlorophyllum molybdites* (MFLU 12-1783). (**a**) basidiospores. (**b**) cheilocystidia. (**c**) pileus covering.

Pileus 70–110 mm, globose to subglobose when young, expanding to convex, plano–convex, with low umbo, with straight margin, with glabrous calotte at center, light-brown to brown (7D5–6), with concolorous patches and squamules scattered toward margin, on concentrically fibrillose background, orange-white (6A2) around umbo, and white around marginal zone; margin fibrillose, fringed, exceeding lamellae when mature. Lamellae free, broad, crowded, white when young, becoming dull green (27D3) when mature, with an eroded edge. Stipe 60–85 × 10–12 mm, widening downward base, 18–23 mm at the base,

white to yellowish-white (4A2), smooth, hollow. Annulus free, descending, with a cuff around stipe, movable, white; underside concolorous with umbo. The context in pileus and stipe white, turning brownish-orange (7C5). Smell carrot-like. Taste not observed. Spore print grayish-green (27C3–4).

Basidiospores [100,5,5] 8.5–13 × 6–7.5 μm, avl × avw = 9.6 × 7.0 μm, Q = 1.3–1.5, avQ= 1.4, ellipsoid-amygdaliform, with truncate apex and germ pore, hyaline or pale-green, thick-walled, dextrinoid, congophilous, cyanophilous, and metachromatic in Cresyl blue. Basidia 29–36 × 10–12 μm, 4-spored, some 1-spored or 2-spored, hyaline, slightly thick-walled. Lamella edge sterile. Cheilocystidia 14–35 × 10–20 μm, clavate, broadly clavate, hyaline, slightly thick-walled. Pileus covering a hymeniderm made up of tightly packed clavate, narrowly clavate, oblong, cylindrical, elements; terminal elements 25–62.5 × 5.0–12 μm, with hyaline to pale-brown parietal pigment. Stipe covering a cutis made up of cylindrical to narrowly cylindrical elements, 4.5–7.5 μm wide, thin-walled, hyaline. Clamp connections not found.

Habitat and distribution: growing in small groups to large groups or fairy rings in grasslands and under shade trees along with grasslands, saprotrophic and terrestrial, throughout northern Thailand.

Material examined: Thailand, Chiang Rai Province, Muang District, Mae Fah Luang University Campus, 7 May 2009, P. Sysouphanthong and J-K. Liu, BJP0041; *ibidem*, 22 July 2012, P. Sysouphanthong, 2012-9 (MFLU 12-1772); *ibidem*, 05 August 2012, P. Sysouphanthong, 2012-12 (MFLU 12-1775); Chiang Rai Province, Muang District, Pongphrabath Village, 17 July 2012, P. Sysouphanthong, 2012-2 (MFLU 12-1765); Pha Yao Province, Muang District, Kuawn Pra Yuan Area, 12 June 2011, P. Callac and S.C. Karunarathna, 2011-11 (MFLU 12-1819).

Notes: *Chlorophyllum molybdites* are easy to recognize by the free and dull green lamellae when fully mature, the grayish-green spore print, and the large basidiomata with a light-pileus with brown squamules at the center. All parts of the basidioma turn orange to red when damaged. It is often found in lawns and grasslands. The species might be confused with *Macrolepiota* species when samples are collected in a young stage. It could also be confused with *Clarkeinda trachodes* (Berk.) Sing, which has a yellow to olive-brown spore print, a much flimsier annulus, and white velar remnants over the pileus scales. *Cl. trachodes* grows in northern Thailand in the same area and habitat as *C. molybdites*, but has not been found outside Asia yet. For differences with *C. globosum*, see notes under that species.

*Chlorophyllum molybdites* is widespread, especially in subtropical and tropical regions [10,14,27,49,52]. In northern-temperate regions in Europe, it was found only indoors [53], Watling [54]. It was first recorded in Thailand by Høiland and Schumacher (1982), and since then, there have been several reports of the species throughout Thailand [17,18]. However, those studies were only based on their macro-morphological characters. This is the first study on *C. molybdites* in Thailand, providing a full description and illustrations. Læssøe et al. [55] recorded this species from Xiangkhouang Province of northern Laos.

In the nrITS sequence analysis (Figure 2) and in the combined nrITS-rnLSU-*rpb2* analysis (Figure 3), three sequences of Thai specimens from this study are clustered with other specimens from other countries worldwide.

*Chlorophyllum molybdites* cause gastrointestinal distress, but the severity of the reactions to this species differs among people and whether the mushrooms have been cooked or not.

The list of synonyms as given above is based on morphology; it is possible that some of the names, such as *Agaricus congolensis* Beeli, in fact, refer to *C. globosum*, which is morphologically very similar to *C. molybdites* [56,57].

3.2.2. Clarkeinda Kuntze

*Clarkeinda trachodes* (Berk.) Singer, Lilloa 22: 413 (1951) [1949]

≡ *Agaricus trachodes* Berk., London J. Bot. 6: 487 bis (1847)
≡ *Chitoniella trachodes* (Berk.) Petch, Ann. R. bot. Gdns Peradeniya 4 (6): 396 (1909)
≡ *Fungus trachodes* (Berk.) Kuntze, Revis. gen. pl. (Leipzig) 3 (3): 480 (1898).
Index Fungorum number: IF 294937; Facesoffungi number: FoF 07066; Figures 13 and 14.

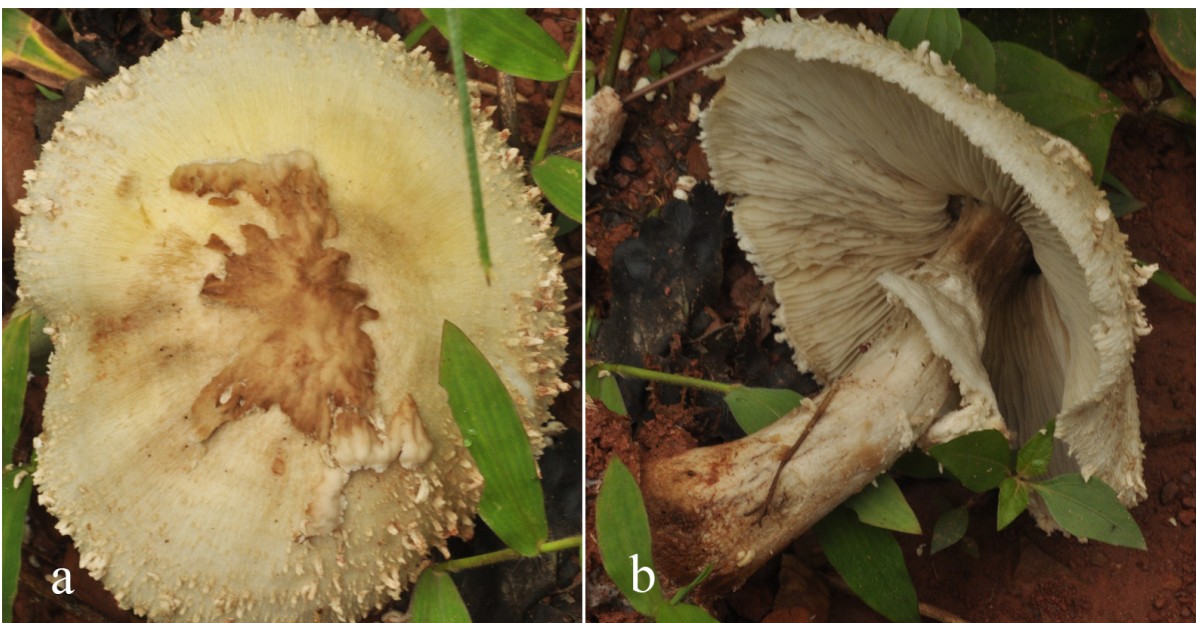

**Figure 13.** Fresh basidiomata of *Clarkeinda trachodes* in situ. (**a**,**b**) MFLU 19-2351.

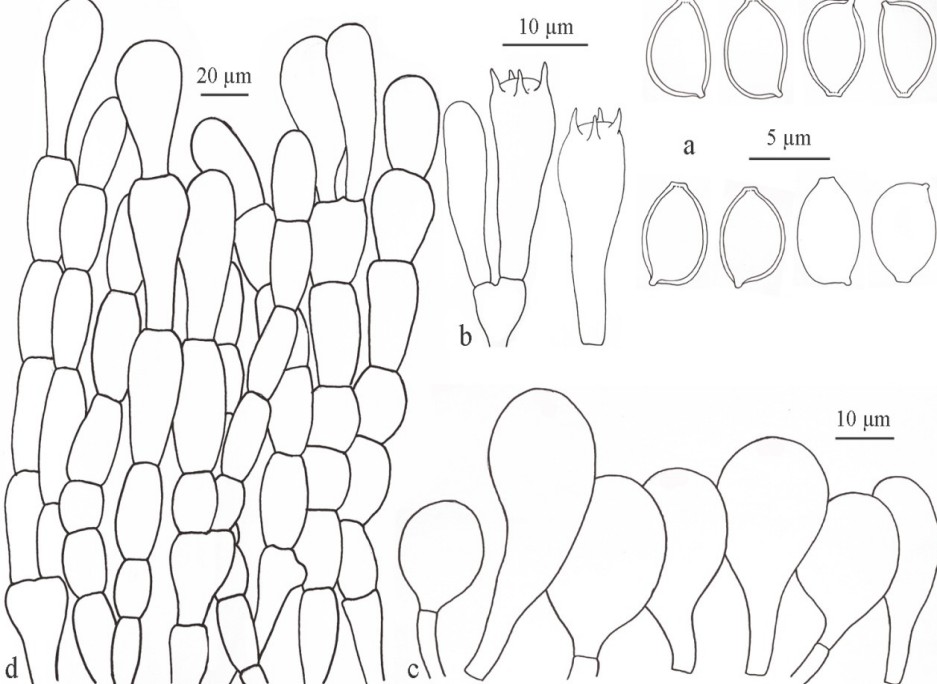

**Figure 14.** Microcharacters of *Clarkeinda trachodes* (MFLU 192-351). (**a**) basidiospores. (**b**) basidia. (**c**) cheilocystidia. (**d**) pileus covering.

Pileus 50–220 mm, first hemispherical to campanulate, expanding to convex or umbonate, applanate to plano-concave, with straight or uplifted margin, with a large irregular patch at the center covering up to 1/3 of pileus, smooth or glabrous, light-brown (6D5–8) to brown (6E7-8), with yellowish-white or pale-yellow (4A3–4) to light-brown (6D5–8)

squamules around patch toward the margin, on yellowish-white or pale-yellow (4A3–4) fibrillose background, with white velar remnants on top the central patch; margin split, with concolorous squamules same as on rest of pileus. Lamellae L = around 180, l = 1–3, 7–12 mm wide, free and remote (2.5 mm) from the stipe, white when young, becoming dull yellow (3B2) to greyish-yellow (4B2) and finally greyish-yellow (3D4) to olive (3C4), with a concolorous or white eroded edge. Stipe 80–150 × 10–28 mm (apex) × 15–28 mm (base), wider at bulbous up to 25 mm wide base, tapering to apex, with white to yellowish-white or pale-yellow (4A3–4) surface, with concolorous squamules same as those on pileus, turning reddish-brown (8E5–7) when touched, hollow. Annulus up to 6 cm wide, attached at the upper part of the stipe, membranous, with eroded margin, fragile and ragged with age; upper side smooth or fibrillose, white to pale-yellow (4A3-4); underside covered with concolorous squamules as on pileus or stipe. Volva present, white. Context white and up to 8 mm thick in pileus with turning orange or reddish-brown (8E5–7), white and hollow in stipe with turning reddish-brown (8E5–7). Odor strong, a mixture of artificial fruit flavor and rubber. Taste not tried. Spore print greyish-yellow (3D4) to olive (3C4) or olivaceous yellow.

Basidiospores [25,1,1] 5.0–6.0 × 3.5–4.0 μm, avl × avw = 5.36× 3.74 μm, Q = 1.32–1.57, avQ = 1.43, ellipsoid ovoid in side-view, broadly to ellipsoid ovoid in frontal view, with truncate apex and germ pore, hyaline, thick-walled. Basidia 19–28 × 5–8 μm, clavate, hyaline, thin-walled, 4-spored. Pleurocystidia absent. Cheilocystidia abundant, 20–50 × 12–35 μm, clavate to broadly clavate, often subglobose to sphaeropedunculate with a short or long stalk, hyaline, slightly thick-walled. Pileus covering in squamules an epithelium composed of 4–5 layers of clavate or short clavate elements, often oblong to fusiform, rarely subclavate in the upper layer, 30–50 × 15–25 μm; subglobose, oblong to clavate in the lower layer, 18–45 × 15–25 μm; elements, with pale-brown to brown partial and intracellular pigment. Stipe covering in squamules same as those on pileus, stipe surface not observed. Clamp connections absent.

Habitat and distribution: growing solitary or in small groups, saprotrophic, on nutrient-rich soil, in the deciduous rain forest of northern Thailand.

Material examined: Thailand, Chiang Mai Province, Mae Taeng District, Pha Deng Village, alt. 900–950 m, 15 August 2011, P. Sysouphanthong, PS2011-12 (MFLU 19-2351); *ibidem*, 22 August 2007, P. Sysouphanthong (MFLU 09-0059); *ibidem*, 25 June 2007, E.C. Vellinga 3550 (UC); *ibidem*, 4 July 2008, E.C. Vellinga 3838 (MFLU 08-1227). Laos, Oudomxay Province, Xay District, Houay Houm Village, 16 August 2014, PS2014-752 (HNL502423).

Notes: *Clarkeinda trachodes* is commonly found in tropical regions of eastern and southern Asia. It has been recorded in Thailand before [2,17,55] and Laos [55]. This study provides a full description, line drawing, and color photos taken in the field. A new nrITS sequence data analysis showed that our new collection clustered together with specimens collected from Thailand [2], Bangladesh [21], and the type location in Sri [46] (Figure 1).

3.2.3. Macrolepiota Singer

*Macrolepiota detersa* Z. W. Ge, Zhu. L. Yang and Vellinga, Fungal Diversity 45: 83 (2010)

Index Fungorum number: IF 518349; Facesoffungi number: FoF 09698; Figures 15 and 16.

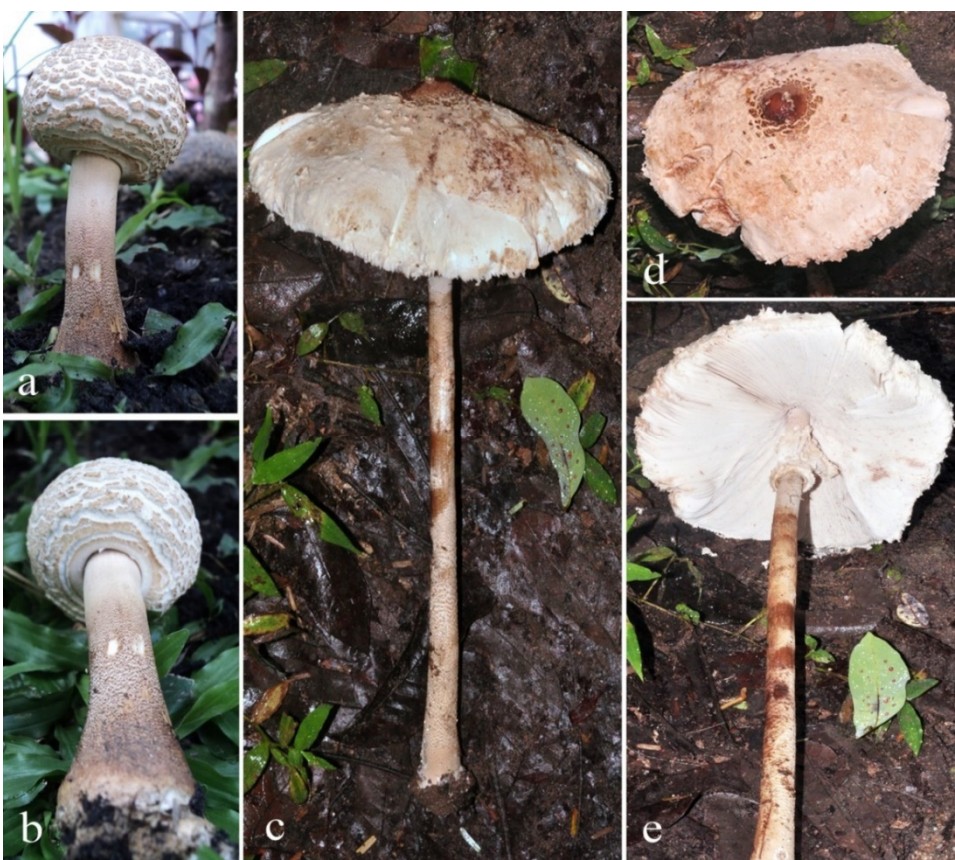

**Figure 15.** Fresh basidiomata of *Macrolepiota detersa* in situ. (**a**,**b**) MFLU 12-1784. (**c–e**) MFLU 12-1772.

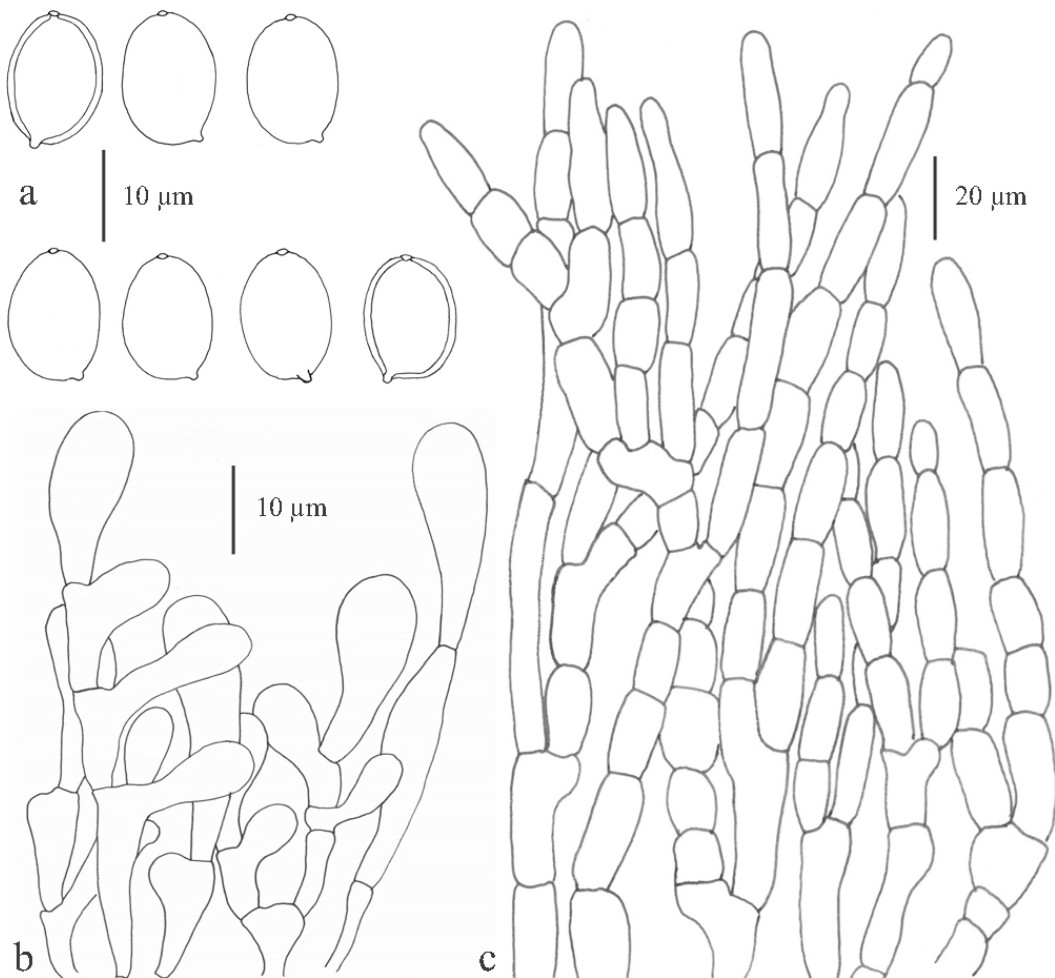

**Figure 16.** Microcharacters of *Macrolepiota detersa* (MFLU 192-351). (**a**) basidiospores. (**b**) cheilocystidia. (**c**) pileus covering.

Pileus 60–160 mm, when young globose to subglobose, expanding to convex, umbonate, broadly campanulate, plane-umbonate, with straight or decurrent margin; when young glabrous, brownish-orange to light-brown or dark red-brown (7C5-6, 8-9 F7-8), surface-breaking around the center, with concolorous squamules or patches around glabrous calotte at the center toward the margin, crowded, with distantly fragile patches toward margin), with concolorous brownish-orange (7C5-6) fibrillose squamules or squamules between patches towards the margin, on white to yellowish-white (4A2) matted-fibrillose to the felted background; margin splitted, covered with light-grey (6C3) to grey orange (6B3) granular-floccose Universal Veil elements. Lamellae free, slightly remote from the stipe, white, sometimes with reddish spots, ventricose to broad, 8-16 mm wide, close to crowded, with the eroded edge. Stipe 80–400 × 12–18 mm, cylindrical with slight wider downward base, with bulb at base, 20–30 mm; dull, dry, scurfy to squamulose overall, minute squamules at apex zone to middle and squamules downward base, concolorous with those squamules on pileus, on brown-grey (6C3) background. Annulus superior, membranous, thin, descending, double, persistent but easily removable, pendulous, 30–40 mm long, with concolorous squamules underside of internal cuff-liked. Context white and thick in pileus; stipe white and hollow, fibrous, turning pinkish-grey when cut, not changing color in bulb. Taste not observed. Smell strong, pleasant. Spore print white.

Basidiospores [50,2,2] 14–16.2 × 9.5–11 μm, avl × avw = 15.2 x 10.1 μm, Q = 1.45–1.62, avQ = 1.52, ellipsoid amygdaliform, ellipsoid to oblong in frontal view, with apical germ pore and hyaline cap, hyaline and thick-walled, dextrinoid, congophilous, cyanophilous, metachromatic in Cresyl Blue. Basidia 38–41 × 12–14 μm, clavate, slightly thick-walled,

hyaline, 4-spored. Lamella edge sterile. Cheilocystidia 20–25 × 7–14 µm, short clavate to clavate, irregular clavate, rarely utriform, sometime branched, hyaline, thin-walled. Pleurocystidia absent. Pileus covering a hymenoderm made up of layers of oblong to cylindrical elements, branched, terminal cylindrical elements cylindrical or fusiform, oblong, with rounded or attenuate to apex, 10–80 × 7–16 µm, slightly thick-walled, with hyaline to pale-brown parietal pigment. Stipe covering not observed. Clamp-connections not found.

Habitat and distribution: solitary, saprotrophic, and terrestrial, growing on the ground beside a grassland or yard, and ground with decomposing deciduous leaves from the mixed rain forest, dominant species being *Castanopsis* spp. in northern Thailand and in northern Laos. This is proposed as a new record for tropical Southeast Asia.

Material examined: Thailand, Chiang Rai, Muang District, Pong Phra Bath Village, 8 June 2011, P. Sysouphanthong, 2011-10 (MFLU 12-1772); Chiang Rai, Mae Fah Luang District, Mae Sa Long Nok Village, 9 July 2012, P. Sysouphanthong, 20112-19 (MFLU 12-1784). Laos, Oudomxay Province, Xay District, Houay Houm Village, 28 July 2014, P. Sysouphanthong, PS2014-567 (HNL502238).

Notes: *Macrolepiota detersa* is recognized by medium to large and very tall basidiomata, white to yellowish-white pileus covering with brownish-orange to light-brown central calotte and patches or squamules, white and free lamellae, long stipe with a draping white membranous annulus attached at the upper part, amygdaliform basidiospores with germ pore and hyaline cap, clavate to narrowly clavate cheilocystidia, and the hymenidermal pileus covering made up of oblong to cylindrical elements.

*Macrolepiota detersa* was originally described, from a temperate region in the Anhui province of eastern China, as an edible species by Ge et al. [58]. It was already known from Japan (as an undescribed species; Vellinga et al. [13] and was later reported from South Korea [59]. Here we record its presence in northern Thailand and Laos [55].

The ITS sequence of the Thai collection is identical to materials from China, Japan, and South Korea with 100% bootstrap support, and it is sister to *M. excelsa* (Figure 4).

*Macrolepiota dolichaula* (Berk. and Broome) Pegler and Rayner, Kew Bull. 23: 365 (1969)

≡ Agaricus dolichaulus Berk. and Broome, Trans. linn. Soc., Bot. 27: 150 (1870)

≡ *Lepiota dolichaula* (Berk. and Broome) Sacc., Syll. Fung. 5: 32 (1887)

≡ *Leucocoprinus dolichaulus* (Berk. and Broome) Pat., Bull. trimest. Soc. mycol. Fr. 29: 215 (1913)

= *Agaricus beckleri* Berk., J. linn. Soc., Bot. 13: 156 (1872)

= *Lepiota beckleri* (Berk.). Sacc., Syll. Fung. 5: 56 (1887)

= *Agaricus stenophyllus* Cooke and Massee, Grevillea 15: 98 (1887)

= *Lepiota stenophylla* (Cooke and Massee) Sacc., Syll. Fung. 9: 4 (1891)

= *Leucocoprinus dolichaulus* var. *cryptocyclus* Pat., Bull. trimest. Soc. mycol. Fr. 29: 215 (1913)

= *Lepiota dolichaula* var. *cryptocycla* (Pat.) Sacc. and Trotter, Syll. Fung. 23: 23 (1925.)

Index Fungorum number: IF 333540; Facesoffungi number: FoF 09699; Figures 17 and 18.

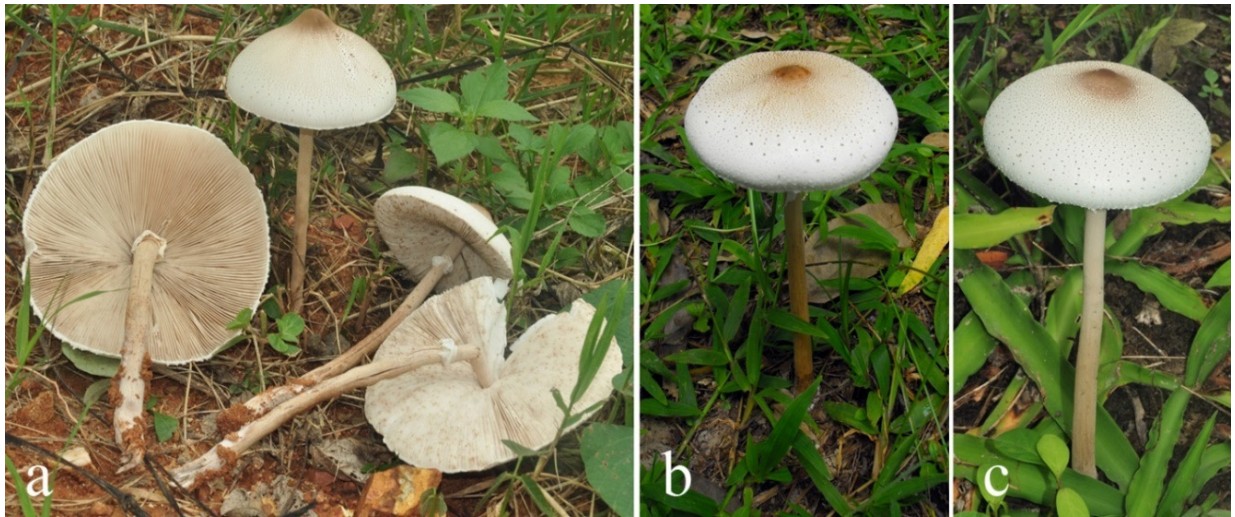

**Figure 17.** Fresh basidiomata of *Macrolepiota dolichaula* in situ. (**a**) MFLU 12-1816. (**b**) MFLU 12-1776. (**c**) HNL503204.

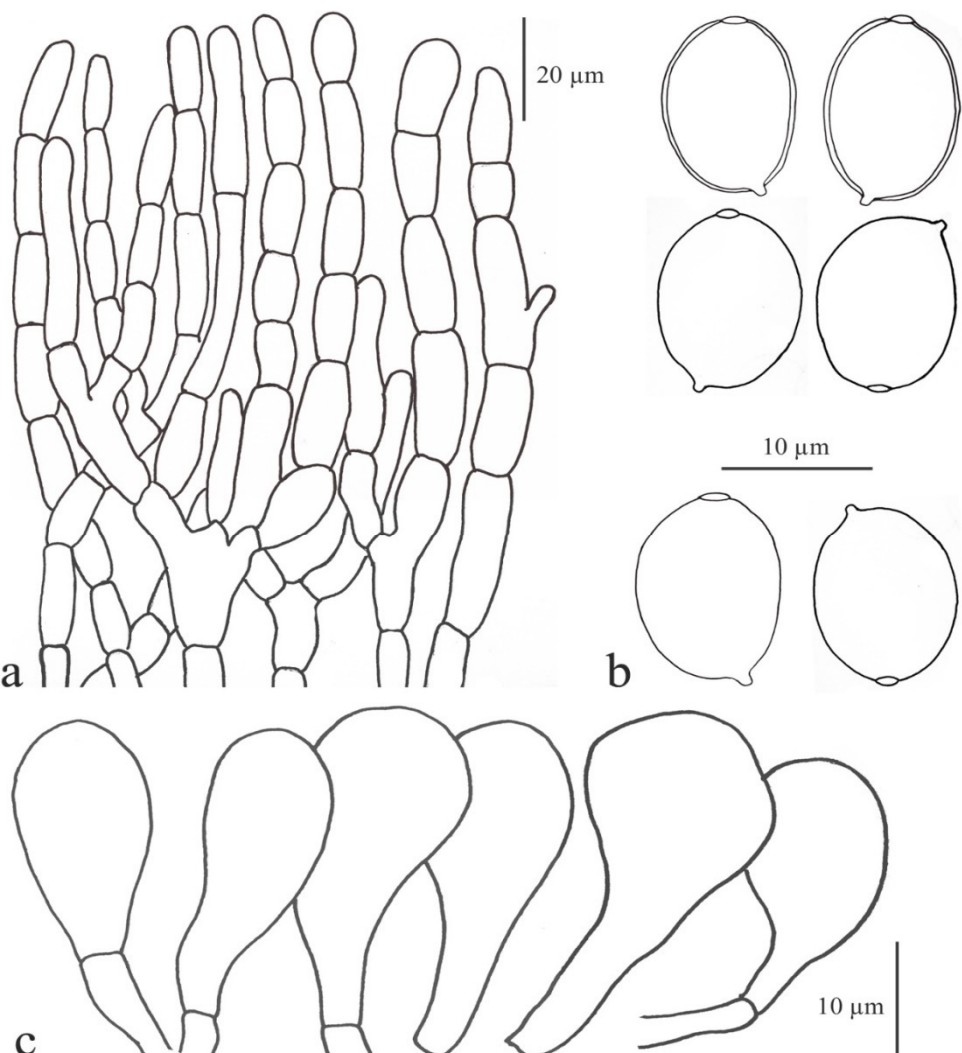

**Figure 18.** Microcharacters of *Macrolepiota dolichaula* (MFLU 12-1816). (**a**) pileus covering. (**b**) basidiospores. (**c**) cheilocystidia.

Pileus 75–175 mm, first subglobose to globose, soon expanding to campanulate, umbonate with high umbo, with inflexed margin, with brownish-orange to light-brown (6C5-6, 6D6-7) glabrous surface and smooth when young, soon surface radially broken from

margin to center, with light-brown (6C5-6, 6D6-7) radially crowded squamules around central calotte toward margin; squamules fragile and distant or completely disappearing when mature, on white fibrillose background; marginal zone with white fibrils, sulcate when mature. Lamellae free, white, crowded, ventricose, wider at pileus margin and narrow to pileus center, 20–28 mm wide. Stipe 140–260 × 10–15 mm, cylindrical, white background, white and smooth from apex to the middle, covered with orange-white to pale-orange (5A2-3) minute squamules downward base, turning pale-red to pastel (7A3-5) when touched, with crowded white rhizomorphs at the base. Annulus membranous, descendent, white on the upper side, with concolorous squamules underside as those on pileus, moveable when mature. Context, white and moderately thick in pileus, white and hollow in the stipe, slowly turning pale-red to pastel (7A3-5) in both pileus and stipe. Taste and smell note observed. Spore print whitish to white.

Basidiospores [100,4,4] 12.5–15 × 8.5–11.5 μm, avl × avw = 13.86 × 10.2 μm, Q = 1.31–1.5, avQ = 1.37, ellipsoid-ovoid, ellipsoid, to ellipsoid-amygdaliform in the side view, ellipsoid in frontal view, hyaline and thick-walled, with germ pore and hyalinous cap over germ pore, dextrinoid, congophilous, cyanophilous, metachromatic. Basidia 30–45 × 10–15 μm, clavate, hyaline, and thin-walled, 4-spored. Lamella edge sterile. Cheilocystidia 20–30 × 10–14 μm, clavate, rarely spheropedunculate, slightly thick-walled, hyaline. Pleurocystidia absent. Pileus covering a hymeniderm made up of oblong to cylindrical elements, terminal elements oblong, cylindrical, short clavate, with rounded or attenuate apex, 15–55 × 6–9 μm, hyaline to pale-brown parietal pigment. Stipe covering not observed. Clamp-connections are present and abundant at the base of basidia and cheilocystidia.

Habitat and distribution: saprotrophic and terrestrial growing in grasslands, under shade trees beside grassland, in pine forests, and on cow dung; solitary or with many basidiomata spread over a wide area; widely distributed and found throughout northern Thailand.

Material examined: Thailand, Chiang Rai, Muang District, Mae Fah Luang University Campus: 20°03′16.9″ N; 99°53′42.6″ E, alt., 410 m, 7 August 2011, P. Sysouphanthong, 2011-15 (MFLU 12-1816); Chiang Rai, Muang District, Pong Phra Bath Village, 15 July 2012, P. Sysouphanthong, 2012-1 (MFLU 12-1764); *ibidem*, 20 August 2012, P. Sysouphanthong, 2012-13 (MFLU 12-1776); *ibidem*, 27 August 2012, P. Sysouphanthong, 2012-15 (MFLU 12-1778); Chiang Rai, Muang District, Mae Pu Ka Village, 14 June 2009, K. Anchalee, BJP020, culture (MFLUCC 10-0366); *ibidem*, 9 July 2009, K. Anchalee, BJP0046 (MFLU 12-1782); Chiang Rai, Mae Fah Luang District, Mae Salong Nok Village: 20°09′121.7′′N; 99°39′26.6″ E, alt., 985 m, 22 July 2012, P. Sysouphanthong, 2012-8 (MFLU 12-1771); Chiang Mai, Mae Taeng District, Pha Deng Village: 19°06′49.1″ N; 98°43′33.5″ E, alt., 971 m, 16 June 2009, P. Sysouphanthong, BJP017, culture (MFLUCC 10-0363). Laos, Xiang Khouang Province, Pek District, Mak Khay Village, 8 June 2016, P. Sysouphanthong, PS2016-81 (HNL503204); Champasak Province, Pak Song District, 3 October 2015, P. Sysouphanthong, PS50 (HNL503277).

Notes: All specimens of *M. dolichaula* are characterized by large basidiomata, a pileus with small, brownish-orange to light-brown, crowded squamules on a white fibrillose background, a long stipe with pale-orange minute squamules, and membranous annulus attached at the upper part of stipe, ellipsoid to ellipsoid-amydaliform spores with germ pore and cap, clavate cheilocystidia, and oblong to cylindrical elements in the pileus covering.

This species is widespread all through tropical Asia and has been reported from Sri Lanka [60–63], India [64], and Vietnam [65], both tropical and temperate China [58,66], Laos [55,67], and is also known from Australia [27,68,69] (Figure 4).

*Macrolepiota dolichaula* is common in Thailand [17], and it was also recorded from Laos by Læssøe et al. [55], however, these studies were based on macromorphology only. Thus, we redescribe this species based on morphology and molecular evidence. In Figure 4, the nrITS sequences from Thailand and Laos group with specimens from China and

Australia with high bootstrap support. *Macrolepiota aberdarensis* Mbaluto and Otieno, a close relative from Kenya, does not differ much from *M. dolichaula* in morphology and nrITS sequences analysis [70]. However, a multiple genes analysis is required to verify both species. It is well possible that the species recorded by Pegler and Rayner [71] from East Africa is *M. aberdarensis*, and not *M. dolichaula* proper. One should also note that *M. abararensis* was not validly published as a herbarium for the type collections was not indicated.

*Macrolepiota excelsa* Vellinga, Sysouph., Thongkl. and K.D. Hyde sp. nov.

*Mycobank umbero*: MB 838685; Facesoffungi number: FoF 09700; Figures 19 and 20.

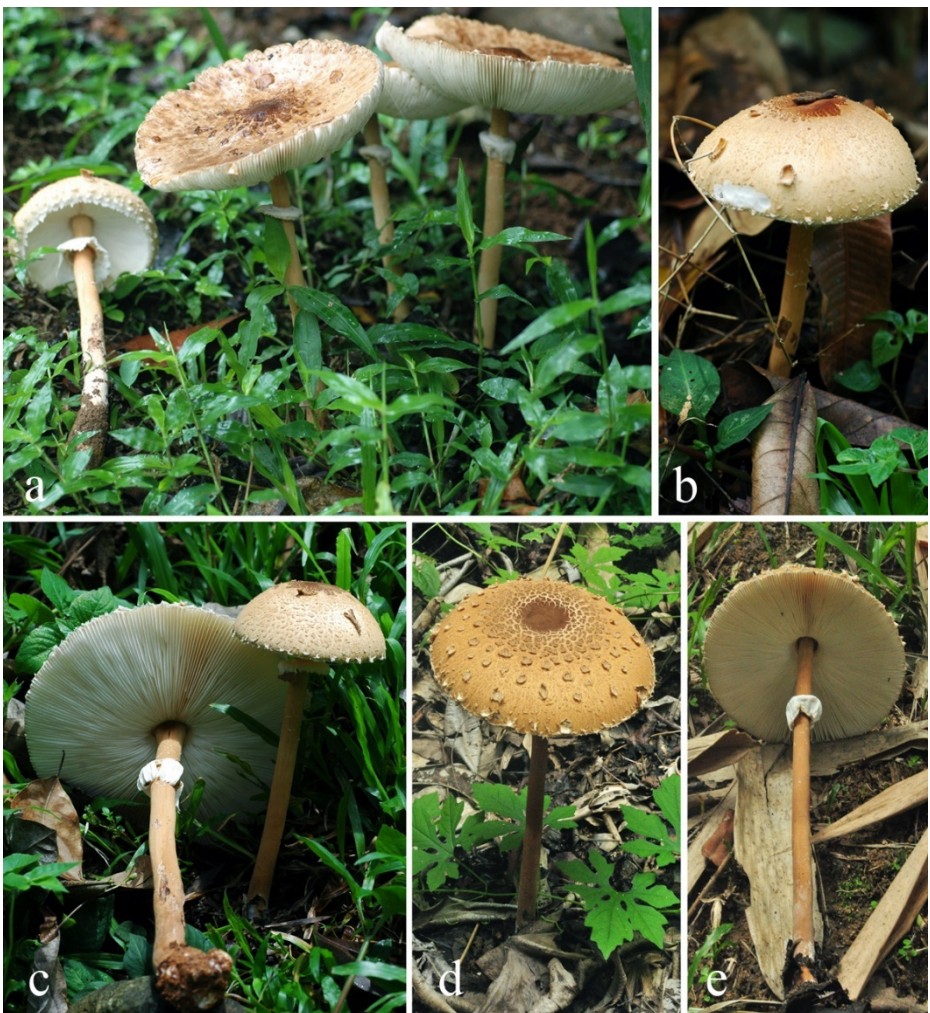

**Figure 19.** Fresh basidiomata of *Macrolepiota excelsa* in situ. (**a**) MFLU 09-0203. (**b**) ecv3553. (**c**) ecv3572 (type). (**d**,**e**) HNL501921.

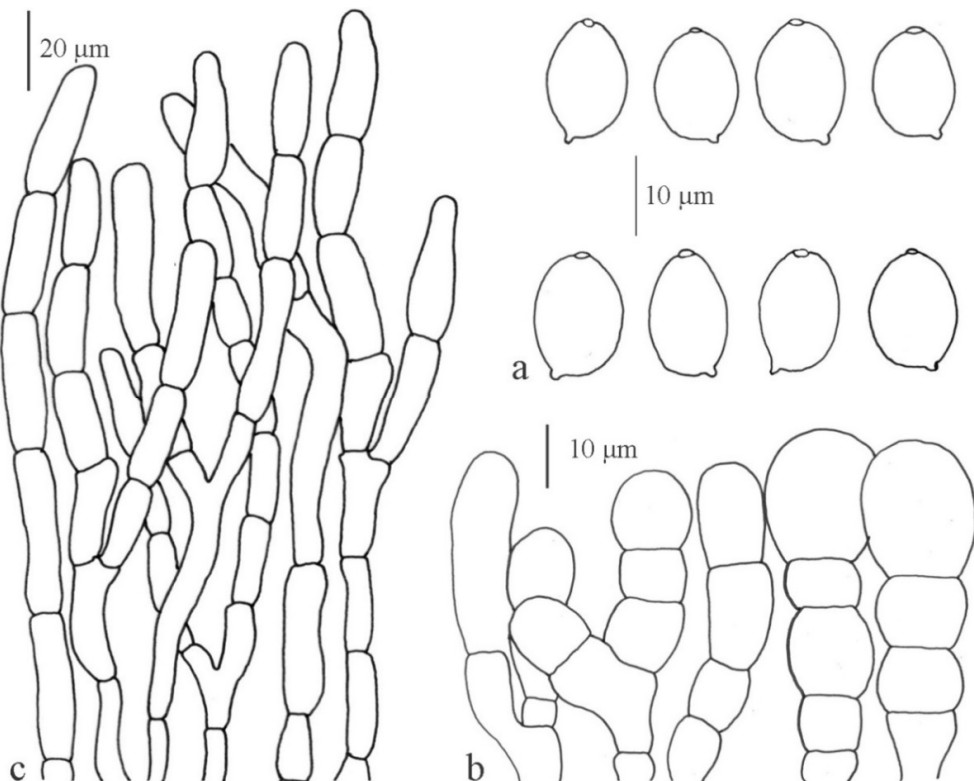

**Figure 20.** Microcharacters of *Macrolepiota excelsa* (ecv3572, Type). (**a**) basidiospores. (**b**) Cheilocystidia. (**c**) pileus covering.

Etymology: The Latin name '*excelsa*' refers to 'high, lofty, distinguished.'

Diagnosis: *Macrolepiota excelsa* is recognized by large brown to dark brown pileus, free and white lamellae, brown to dark brown cylindrical stipe with the white membranous annulus, broadly ellipsoid to ellipsoid basidiospore with a germ pore, utriform to narrowly clavate cheilocystidia, a hymeniderm made up of clavate to cylindrical elements in pileus covering.

Holotype: ecv3572 (UC).

Pileus 120–220 mm, when young globose to subglobose, paraboloid, expanding to campanulate, convex, umbonate or not, sometimes slightly depressed at the center, when young completely brown (6E5), later covering broken and with dark brown, irregular calotte (6F4-8; Mu. 7.5 YR 5/4, 6/6) at the center, with concolorous big patches and squamules around calotte toward the margin, patches fragile when mature; background radially fibrillose and light to greyish-brown (6D3-5; 10 YR 7/4–7.5 YR 8/6–4), slightly squarrose especially in the outer part, where the fibrils are more spread out and show the white background in between; margin irregular and exceeding lamellae. Lamellae, L = 80–130, l = 0 or 1, moderately crowded to crowded, free and remote (up to 4 mm) from stipe, off-white, broad and ventricose, up to 12 mm wide, with white floccose (cystidiose) edge. Stipe 130–350 × 7–10 mm at apex, 10–18 mm wide in the center, cylindrical or slightly widening downward, 23–30 mm wide at base; base in some specimens bulbous; completely covered in brown tomentose-velvety light-brown to brown, or reddish-brown (6D4-5, 6E5-6, 7.5 YR 7–6/6) covering, which breaks open into small very fine to fine horizontal bands that can form zig-zag patterns, hollow with central white cottony strand, protruding into pileus. Annulus a white descending or ascending cuff, and a flaring double part, white to yellowish-white (4A2) membranous on the upper side with a fringed edge; underside with squamules as on pileus; movable with age. Context thick and soft, white and dull in pileus; in stipe buff and hollow. Smell fungoid. Taste not observed. Spore print white.

Basidiospores [90,3,3] 11.8–16.5 × 8.3–13.5 μm, avl × avw = 13.2–15.3 × 9.2–11.2 μm, Q = 1.2–1.51, avQ = 1.3–1.45, in side-view broadly ellipsoid to ellipsoid, with germ pore covered by the hyaline cap, thick-walled, hyaline to pale-yellow, dextrinoid, congophilous, metachromatic. Basidia 30–40 × 11–19 μm, clavate, slightly thick-walled, 4-spored, rarely 2-spored, with basal clamp-connection. Lamella edge sterile. Cheilocystidia in chains of up to 5 elements; terminal elements 7–40 × 7–30 μm, variable in size and shape, usually subglobose, occasionally clavate, with rarely cylindrical, hyaline. Pleurocystidia not found. Pileus covering at umbo and calotte a hymenoderm made up of hemispherical to cylindrical elements, thick-walled, hyaline to pale-brown, 20–92.5 × 10–12.5 μm. Stipe covering a hymeniderm made up of cylindrical elements and hyphae, 8.7–15 μm wide. Clamp-connections at the base of basidia and cheilocystidia.

Habitat and distribution: growing solitary to small group, saprotrophic, and terrestrial in half-open forests and on the edge of grasslands; known from Chiang Mai province of northern Thailand and Oudomxay province of North Laos.

Material examined: Thailand, Chiang Mai Province, Mae Taeng District, Pha Deng Village, 25 June 2007, E.C. Vellinga 3553 (UC); *ibidem*, 3 July 2007, R. Walleyn (coll. E.C. Vellinga 3599) (UC); *ibidem*, 28 September 2008, P. Sysouphanthong (MFLU 09-0203); *ibidem*, 20 June 2009, P. Sysouphanthong and J.K. Lui, BJP0023, culture (MFLUCC 10-0369); Chiang Mai province, Mae Taeng District, Pongduad Village, 22 June 2009, P. Sysouphanthong and J.K. Lui, BJP0029, culture (MFLUCC 10-0375). Laos, Oudomxay Privince, Xay District, Houy Houm Village, 30 June 2014, P. Sysouphanthong, PS2014-260 (HNL501921); *ibidem*, 16 July 2014, P. Sysouphanthong, PS2014-437 (HNL502108); *ibidem*, 05 August 2014, P. Sysouphanthong, PS2014-648 (HNL502319).

Notes: *Macrolepiota excelsa* is a new species found in northern Thailand and northern Laos. Macroscopically this species resembles *M. procera* (Scop.: Fr.) Singer very much in general appearance, but it differs in the brown, not grey-brown, colors on pileus and stipe. The ITS sequence, however, does not place this species in the vicinity of *M. procera*, but closely related to *M. detersa*, *M. dolichaula*, the Australian sequestrate species *M. turbinata* T. Lebel, and *M. aberdarensis* Mbaluto and Otieno from Kenya (Figure 4). *Macrolepiota detersa* differs in brownish-orange to light-brown squamules on pileus and yellowish-white background, the very well-developed hanging annulus, and the branched cheilocystidia (Figures 15 and 16); the holotype from China showed more clavate to broadly clavate to pyriform and rarely subfusiform cheilocystidia [58]. The pileus of *M. dolichaula* is much lighter colored with small brown squamules on a white background, and more clavate to sphaeropedunculate cheilocystidia (Figures 19 and 20). *Macrolepiota aberdarensis*, a species resembling *M. dolichaula,* is distinghuised from *M. detersa* by the lighter color of squamules on pileus and white background and the cheilocystidia not arranged in chains [70]. Another species in the same clade, *Macrolepiota umbonata* H. J. Cho, H. Lee, and Y.W. Lim, described from South Korea differs from *M. excelsa* by the distinct umbo on the pileus, the white background of the pileus surface, and superior and non-membranous annulus [59]

The stipes of *M. excelsa* have a central cord, similar to the stipes of *Coprinus comatus* (O.F. Müll.: Fr.) Pers. and *Montagnea arenaria* (DC.) Zeller [72]. This is the first report of this feature outside the *Coprinus comatus* clade.

*Macrolepiota velosa* Vellinga and Zhu L. Yang, Mycotaxon 85: 184 (2003).

Index Fungorum number: IF 373847; Facesoffungi number: FoF 09701; Figures 21 and 22.

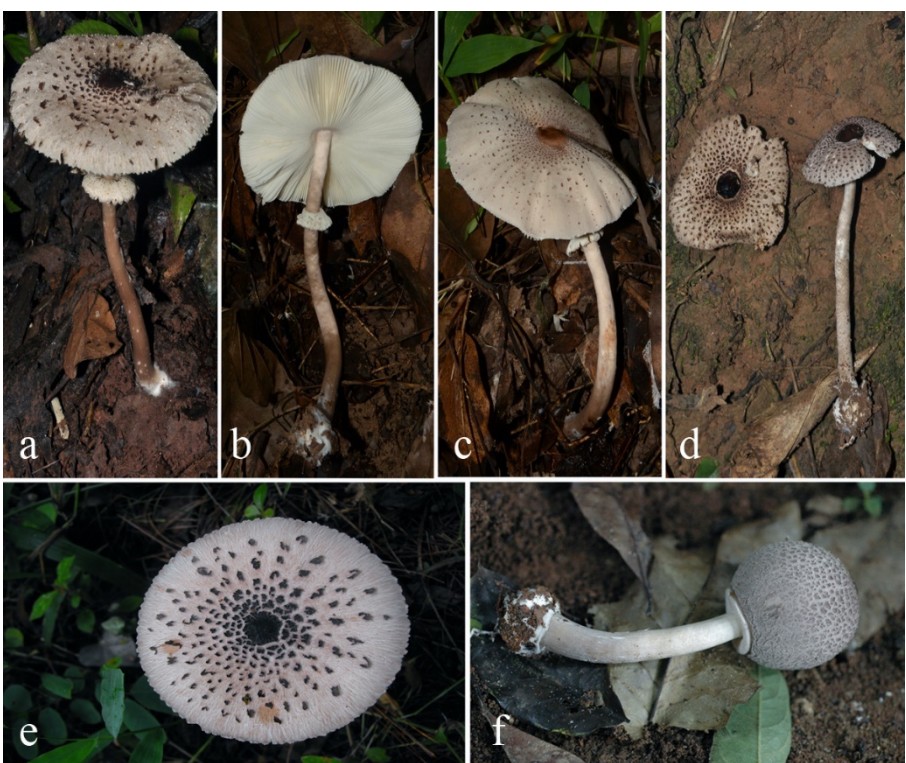

**Figure 21.** Fresh basidiomata of *Macrolepiota velosa* in situ. (**a**,**b**) HNL502655. (**c**) MFLU 09-0052. (**d**) HNL502371. (**e**) MFLU 12-1818. (**f**) MFLU 09-0055.

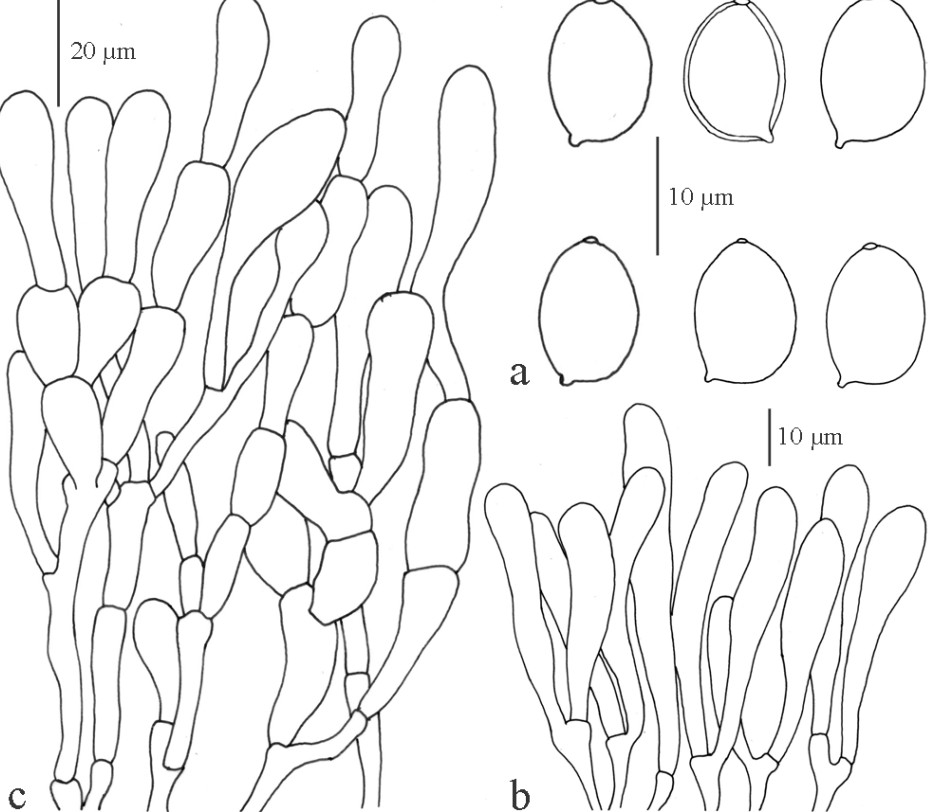

**Figure 22.** Microcharacters of *Macrolepiota velosa* (MFLU 09-0052). (**a**) basidiospores. (**b**) cheilocystidia. (**c**) pileuscovering.

Pileus 100–140 mm, slightly globose when young, expanding to hemispherical, parabolic to campanulate with wide umbo, straight margin, center with a round to star shape or calotte, 10–15 mm diam., with uplifted margin, dark brown (8F4) to dark, with dark brown (8F4) warts or squamules around center toward the margin, squamules long around the marginal zone, with uplift tip, with orange write (5A2) radially fibrillose, background, with white to orange (5A2) white remnant partial veil at the center and scattered on the surface, in mature sample 4–6 grooves, with split margins, exceeding lamellae, fringe. Lamellae free, white, remote from the stipe, close to crowded, ventricose, up to 13 mm wide with white fine-floccose at lamellae surface and marginal edge turning brownish-orange at the edge. Stipe 80–160 x 8–115 mm, long cylindrical, wider at the base, with 20 mm wide bulb, white, smooth, yellowish-white (4A2) fibrillose at apex, with brown (7E4) fibrillose at base zone, with white membranous volva at base. Annulus ascending, movable, with cuff, white to cream at the upper side, grayish-orange (5B3) at underside, thick, with broken brownish beige (6E3) margin. The context in pileus white, 4–6 mm wide, in stipe white. Odor such as vitamin B. Taste mild, sweet.

Basidiospores [75,3,3] 9–11.8 × 6.5–7.5 µm, avl × avw + 9.5 × 7.0 µm, Q = 1.28–1.68, avQ = 1.35, in the side view broadly to ellipsoid-amygdaliform, in frontal view ellipsoid, with germ pore covered by hyaline cap, thick-walled, dextrinoid, congophilous, cyanophilous, metachromatic in Cresyl Blue. Basidia 28–35 × 11–13 µm, 4-spored, clavate, hyaline. Lamella edge sterile, with crowded cheilocystidia. *Cheilocystidia* 20–85 × 5.5–8 µm, cylindrical, wide at apex, with granular content at apex; clamp-connections absent. *Pleurocystida* absent. *Pileus covering* a trichoderm, in upper layer or squamules, made up of clavate to long clavate terminal elements, 20.0–120 × 4–13 µm, with brown parietal and intraocular pigment. *Stipe covering* a cutis composed of hyaline hyphae and cylindrical elements up to 7 µm wide. Clamp-connections absent.

Habitat and distribution: solitary or in small groups, saprotrophic and terrestrial on soil or ground with decomposing leaves mixed with humus; distributed in the rainforest in the highlands in Chiang Mai and Chiang Rai Provinces.

Material examined: Thailand, Chiang Rai Province, Mueng District, Hua Doi Village, 15 August 2009, P. Sysouphanthong and Jainkui Lui, BJP0072, culture (MFLUCC 10-0417); Chiang Rai, Mae Fah Luang District, Pa Kluay Village, 22, August 2011, P. Sysouphanthong, 2011-14 (MFLU 12-1781); Chiang Mai, Mae Taeng District, Pha Deng Village, 12 August 2007, P. Sysouphanthong, PNG055 (MFLU 09-0055); *ibidem*, 12 August 2012, P. Sysouphanthong, PNG052 (MFLU 09-0052), *ibidem*, 19 July 2007, P. Sysouphanthong, PNG015 (MFLU 09-0015). Laos, Oudomxay Province, Xay District, Houay Houm Village, 6 August 2014, P. Sysouphanthong, PS2014-700 (HNL502371); Oudomxay Province, Xay District, Houay Hoom Village, 6 September 2014, P. Sysouphanthong, PS2014-984 (HNL502655); Xekong Province, Tha Taeng District, Dongmakjong Forest, 30 September 2015, P. Sysouphanthong, PS1 (HNL503228).

Notes: *Macrolepiota velosa* differs from other *Macrolepiota* species as it is covered with dark brown to dark squamules on pileus and has a white volva at the basal stipe bulb, cylindrical to narrowly clavate cheilocystidia; the pileus covering is a hymeniderm made up of long, narrowly clavate terminal elements, with pale-brown parietal and intracellular pigment; clamp-connections were not found in the basidiomata. This is also the smallest of the Thai *Macrolepiota* species.

Ge et al. [58] and Vellinga et al. [2] already mentioned the occurrence of *M. velosa* in northern Thailand. [73] Vellinga and Yang (2003) discussed all volvate *Macrolepiota* species. The nrITS sequences of the Thai species are grouped with specimens from China and form a sister group with the volvate *M. eucharis* from Australia (Figure 4). Sysouphanthong et al. [67] recorded *M. velosa* in central Laos, and here we report the species also from northern and southern parts of Laos.

### 3.2.4. Pseudolepiota Z.W. Ge

*Pseudolepiota zangmui* Z.W. Ge in Ge and Yang, Phytotaxa 312: 252 (2017).

Index Fungorum number: IF 820482; Facesoffungi number: FoF 07068; Figures 23 and 24.

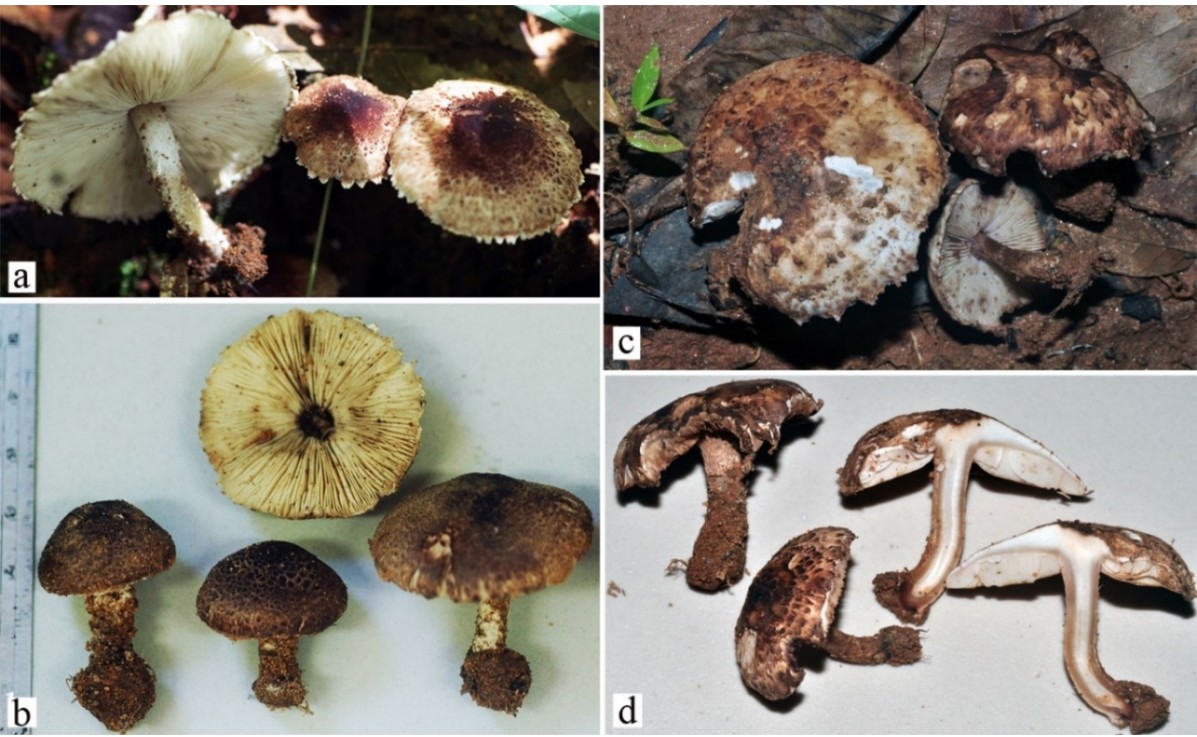

**Figure 23.** Fresh basidiomata of *Pseudolepiota zangmui*. (**a**) MFLU 10-0515. (**b**) MFLU 10-0518. (**c**,**d**) MFLU 192-355.

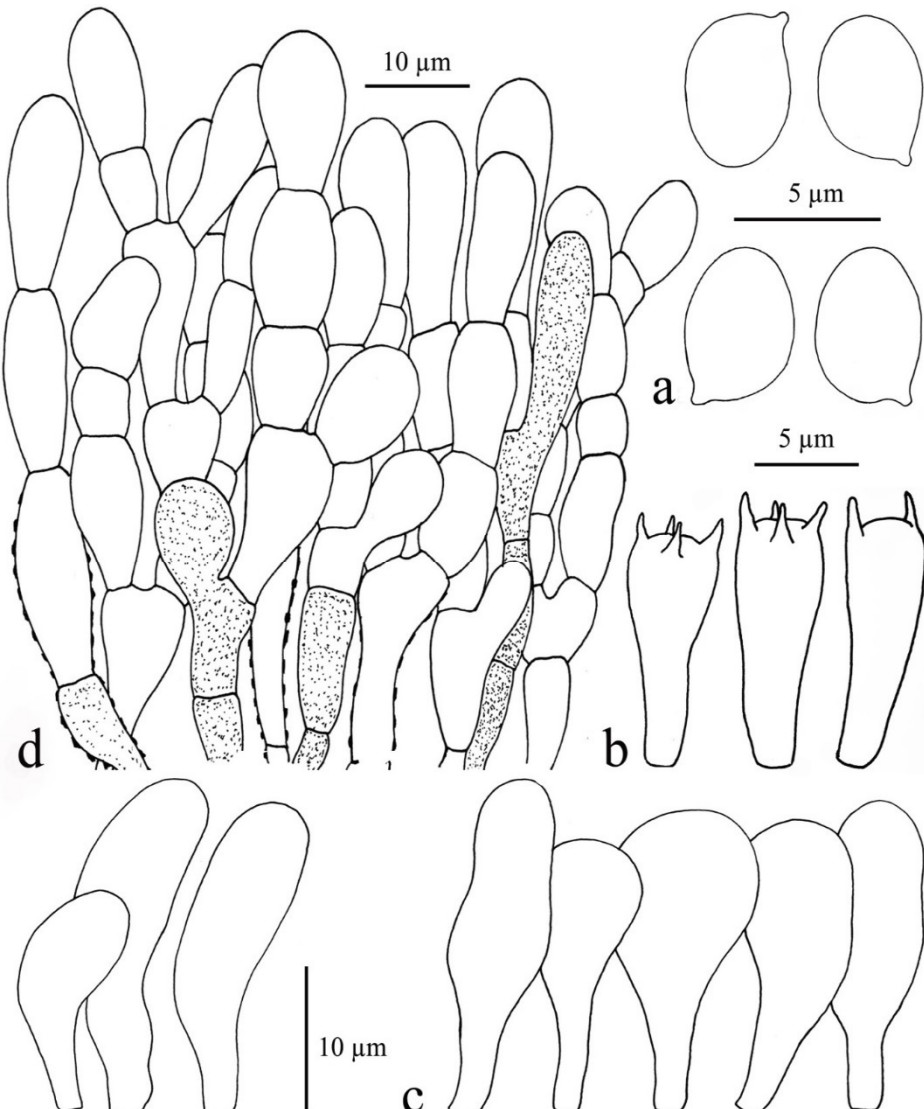

**Figure 24.** Microcharacters of *Pseudolepiota zangmui* (MFLU 10-0515). (**a**) basidiospores. (**b**) basidia. (**c**) cheilocystidia. (**d**) pileus covering.

Pileus 20–50 mm, at first hemispherical, campanulate, expanding to convex, umbonate or often plano-convex with an umbo and inflexed margin; surface covering violet-brown (10F5-8, 11F-8), darker at umbo or center, dark ruby to dark (12F4–8), smooth or rough, later splitting off and becoming irregularly concolorous squamulose, with larger scales around the umbo and with smaller squamules toward the margin, with dull red (9B3–4) fibrillose or fibrillose squamules at the marginal zone on a white to orange-white (6A2) background; squamules are fragile when mature; margin appendiculate, with white to dull red (9B3–4) fibrillose remnants. Lamellae free, at first white, turning yellowish-white when mature and greyish-orange (5B3–4) when dried, 3.5–5 mm wide, crowded, with 4–5 lamellulae in between lamellae, slightly fragile with age, with an eroded edge. Stipe 25–40 × 5–8 mm, equal, slightly wider at base zone, completely covered by white to yellowish-white fibrils, sometimes with white remnants at the annular zone. Annulus a white annular zone. Context white in pileus, up to 4 mm wide, white in the stipe, changing to reddish with age, hollow. Spore print whitish. Taste and smell not observed.

Basidiospores [2,2,50] 4.8–5.2 × 3.5–4.5 µm, avl × avw = 5.0 × 4.0 µm, Q = 1.15–1.37, Qav = 1.25, in side-view broadly ellipsoid to ellipsoid-ovoid, in frontal view broadly ellipsoid to ellipsoid, hyaline to pale-yellow in water and KOH, slightly thick-walled without a germ pore, dextrinoid, congophilous, cyanophilous, not metachromatic. Basidia 15–18 ×

4.8–5.0 µm, clavate, 4-spored, some 2-spored. Lamella edge sterile. Cheilocystidia 15–24 × 6–10 µm, most clavate, some with a long stalk, rarely utriform and sphaeropedunculate, thin-walled, hyaline. Pleurocystidia absent. Pileus covering hymenoderm made up of 3–4 layers of elements, oblong to cylindrical in the lower layer, mostly with clavate, rarely subclavate, elements in the upper layer, 12–30 × 5.0–10 µm, thick-walled, smooth and with hyaline to pale-yellow parietal pigments, with brown intracellular pigments and encrusted walls in some lower elements. Clamp connections absent.

Habitat and distribution: in small to large groups, saprotrophic and terrestrial in deciduous forests dominated by *Lithocarpus* spp. and *Castanopsis* spp. at 650–950 m alt. Found in China and Thailand.

Material examined: THAILAND, Chiang Rai Province, Muang District, Forest of Hua Doi Village, 23 September 2009, P. Sysouphanthong, PHO25 (MFLU 10-0515); *ibidem*, 25 September 2009, P. Sysouphanthong, PHO28 (MFLU 10-0518); Chiang Rai Province, Muang District, Mae Fah Luang University Campus, 13 August 2019, P. Sysouphanthong, PS2019-74 (MFLU 19-2360). Chiang Mai Province, Mae Taeng District, Pha Deng Village, alt. 900–950 m, 7 August 2018, P. Sysouphanthong, PS2018-84 (MFLU 19-2355).

Notes: *Pseudolepiota zangmui* is characterized by squat basidiomata in which the pileus diameter is equal or slightly longer than the stipe length; the pileus is covered in violet-brown to dark ruby squamules with dull red fibrils between squamules, and it has a white appendiculate margin; the basidiospores are pale-yellow, and oblong-ovoid in side-view, the cheilocystidia are clavate, and the pileus covering is hymenidermal.

The type of *Pseudolepiota zangmui* was described from Xishuangbanna, Yunnan Province, Southwest Yunnan, China [59]; it closely resembles *Xanthagaricus* species but differs in the white to pale-cream spore print and the squat basidiomata. Based on phylogenetic analyses of combined ITS and LSU data with a wide sampling of *Xanthagaricus* species, it is closely related and basal to *Xanthagaricus* with good support (Figure 1). However, in a phylogenetic analysis including protein-coding genes [32], *Pseudolepiota* is clearly separated from *Xanthagaricus*.

3.2.5. Xanthagaricus (Heinem.) Little Flower, Hosag. and T.K. Abraham

*Xanthagaricus necopinatus* Iqbal Hosen, T.H. Li, and G.M. Gates in Hosen, Song, Gates, Karunarathna and Li, MycoKeys 28: 9 (2017).

Index Fungorum number: IF 820482; Facesoffungi number: FoF 07068; Figures 25 and 26.

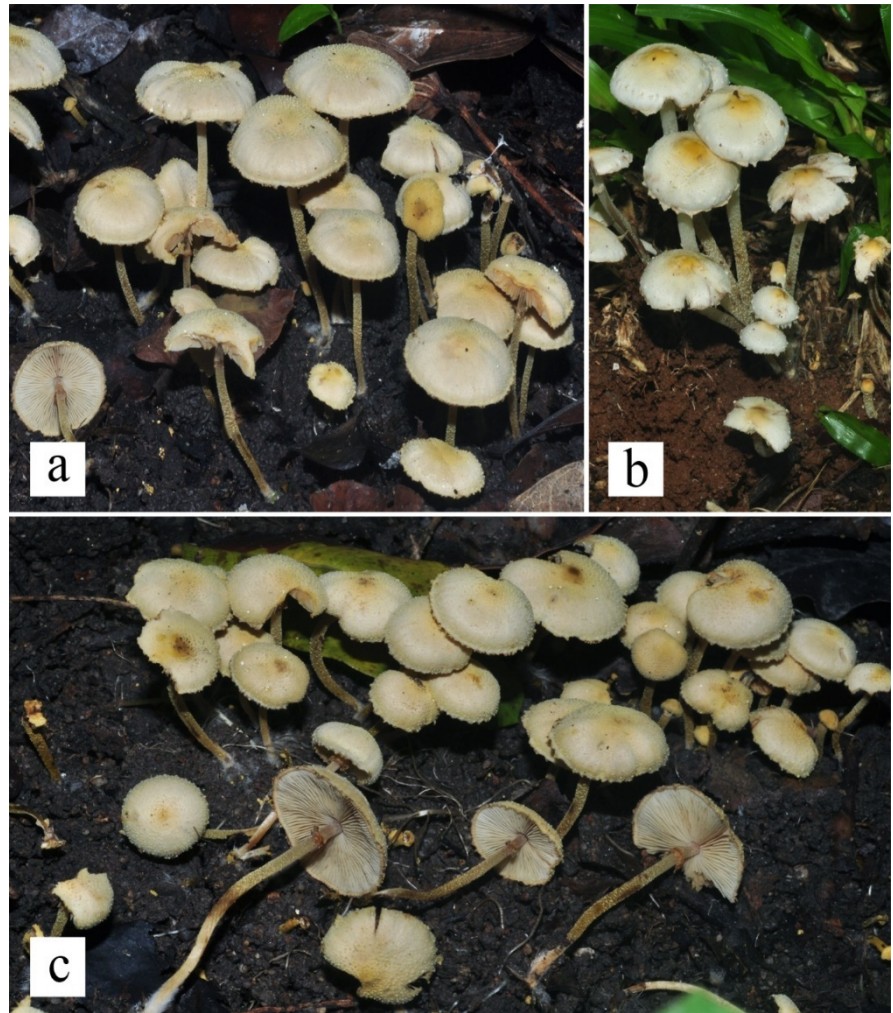

**Figure 25.** Basidiomata of *Xanthagaricus necopinatus* in situ. (**a**) MFLU 19-2353. (**b**) MFLU 19-2359. (**c**) MFLU 19-2358.

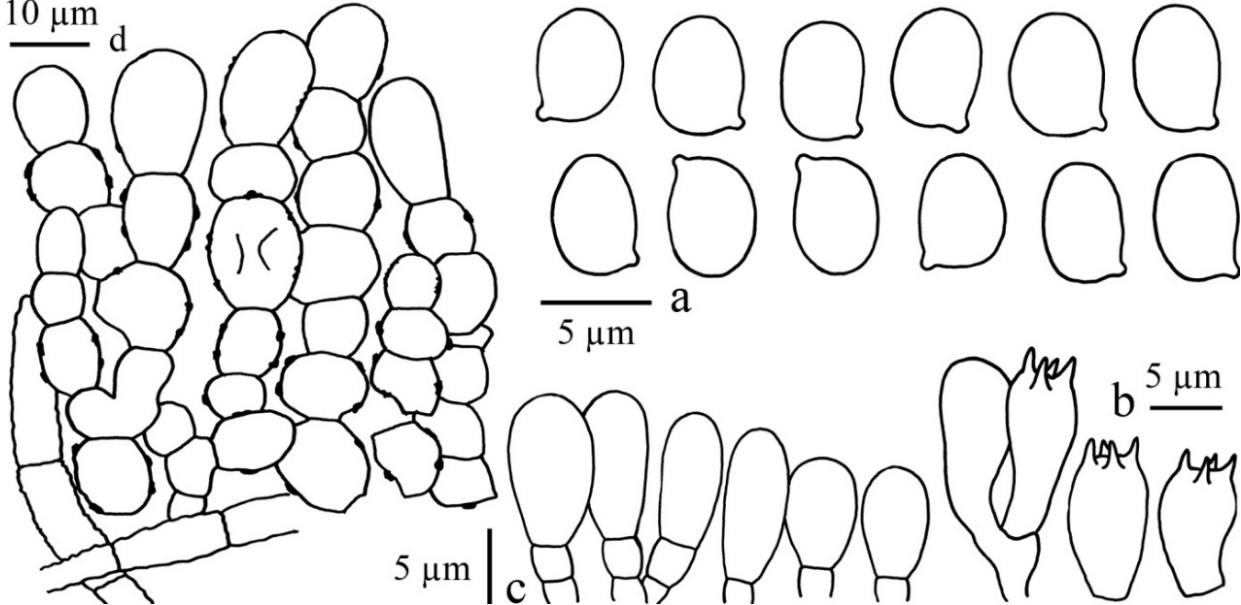

**Figure 26.** Microcharacters of *Xanthagaricus necopinatus* (MFLU 19-2353). (**a**) basidiospores. (**b**) basidia. (**c**) cheilocystidia. (**d**) pileus covering.

Pileus 10–20 mm, first paraboloid, expanding to convex, plano-convex, with or without umbo, with straight or slightly incurved margin; when young covered with crowded squamules, greyish-yellow to yellow (3B3–8), with brownish-yellow squamules (5C7–8) at center; when mature with yellowish-white to yellow (3A2–6) squamules on yellowish-white to pastel yellow (3A2–4) background; margin with concolorous squamules as on pileus surface and with light-orange (6A4-5) appendiculate velar remnants. Lamellae, l = 3 series, free, 2–4 mm wide, yellowish-white to orange-white (4A2, 5A2) when young, becoming orange-grey to greyish-orange (5B2–3) when mature, broadly ventricose, with white, eroded edge. Stipe 20–45 × 2–3 mm, cylindrical, white fibrillose at apex zone, with yellowish-white to yellow (3A2-6) squamules below annulus to base, with orange-grey to greyish-orange (5B2–3) background, turning greyish red (7B4–5). Annulus attached to the apical zone of the stipe, membranous, greyish-orange (6B4–6), sometimes fragile with age. Context white to orange-white (5A2) in pileus, up to 2 mm thick at the center; hollow in stipe and concolorous with the surface. Odor and taste not observed. Spore pint not observed.

Basidiospores [75,3,3] 4.0–5.2 × 2.5–3.5 μm, avl × avw = 4.71 × 2.90 μm, Q = 1.42–1.80, avQ = 1.62, ellipsoid to oblong-ovoid in frontal view, ellipsoid to oblong in side-view, thick-walled, smooth, hyaline to pale-brown, without a germ pore. Basidia 15–20 × 4.5–7 μm, short clavate to clavate, thin-walled, hyaline, 4-spored. Pleurocystidia absent. Cheilocystidia 17–25 × 5–10 μm, short clavate to clavate, often ellipsoid, with 1 or 2 septa at the base, slightly thick-walled, hyaline. Pileus covering an irregular epithelium composed of oblong to short clavate elements in the upper layer, 10–20 × 5–15 μm, with irregular globose to subglobose elements in the lower layer, 5–15 μm wide, slightly thick-walled, rough-walled, encrusted, with pale-brown to brown parietal and intracellular pigments, with hyaline to brown rough-walled and encrusted, up to 8 μm wide hyphae at base of the epithelial layers. Stipe covering an epithelium similar to that on pileus. Clamp connections absent.

Habitat and distribution: growing in large groups, saprotrophic on humus-rich soil with dead leaves and wood under trees or in grassland; commonly found in Chiang Mai and Chiang Rai Provinces, northern Thailand.

Material examined: Thailand, Chiang Mai Province, Muang District, Chiangmai University Campus, 28 July 2018, P. Sysouphanthong, PS2018-56 (MFLU 19-2353); Chiang Rai Province, Muang District, Forest of Mae Fah Luang University Campus, 8 June 2019, P. Sysouphanthong, PS2019-45 (MFL 19-2358); *ibidem*, 8 August 2019, P. Sysouphanthong, PS2019-67 (MFL 19-2359).

Notes: *Xanthagaricus necopinatus* has the slender basidiomata characteristic for the genus, and a pale-yellow pileus and stipe with squamules, velar remnants on the pileus margin, hyaline to pale-brown basidiospores, clavate to ellipsoid cheilocystidia, and an irregular epithelium on pileus and stipe; clamp-connections are absent. The basidiospores are smooth.

*Xanthagaricus necopinatus* was originally described from Bangladesh [34], and this is the first report of this species outside that country. Thai specimens completely resemble the type specimens, but the stipe covering of type specimens is a cutis while the Thai specimens have an epithelium on the stipe. However, the nrITS sequences of Thai specimens are clustered with the type specimens with high BS (Figure 1).

*Xanthagaricus necopinatus* can be confused with *X. flavosquamosus* T.H. Li, Iqbal Hosen and Z.P. Song, described from China, but differs from that species in the smoother pileus and the smooth (vs. rough) basidiospores. The differences in the width and shape of the cheilocystidia as mentioned by Hosen et al. [74], are not supported by the data from the Thai collections.

*Xanthagaricus purpureosquamulosus* Sysouph., Thongkl. and K.D. Hyde sp. nov.

Mycobank umbero: MB 833740; Facesoffungi number: FoF 07071; Figures 27 and 28.

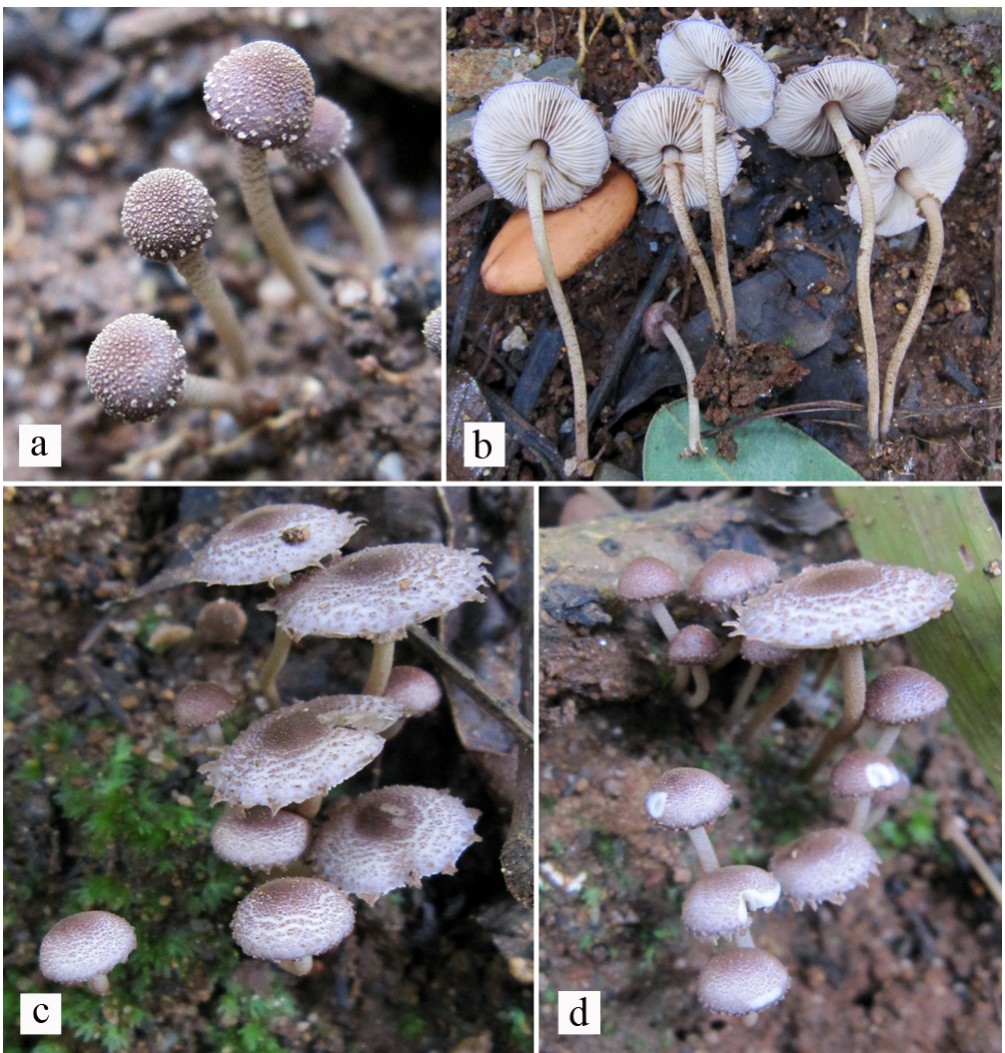

**Figure 27.** Fresh basidiomata of *Xanthagaricus purpureosquamulosus*. (**a**–**c**) MFLU 19-2354 (holotype). (**d**) MFLU 192356.

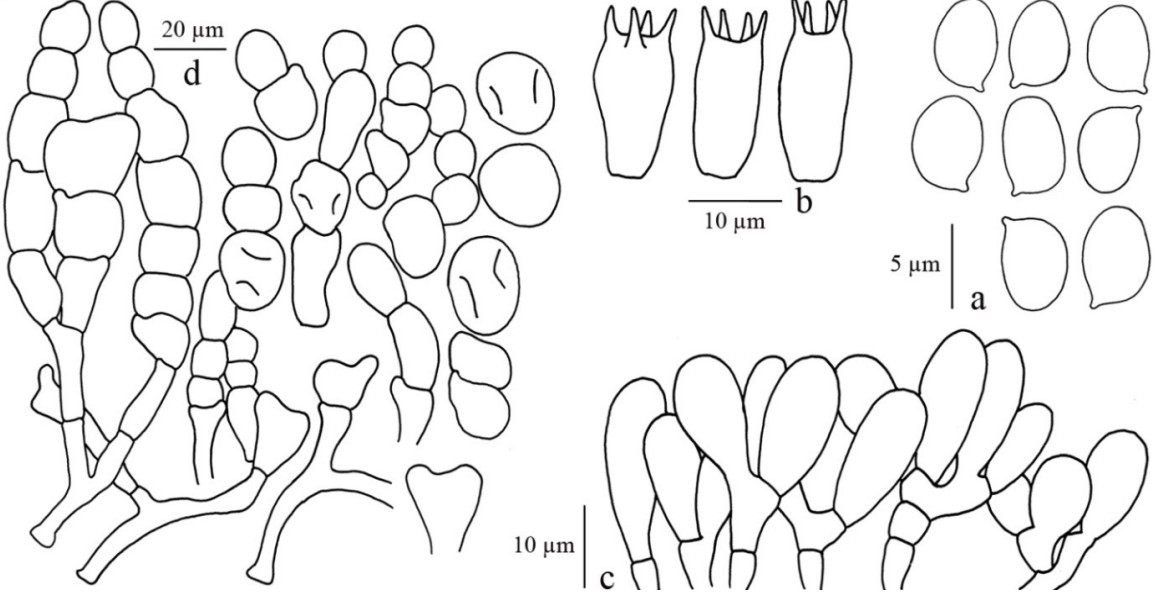

**Figure 28.** Microcharacters of *Xanthagaricus purpureosquamulosus* (MFLU 19-2354, holotype). (**a**) basidiospores. (**b**) basidia. (**c**) cheilocystidia. (**d**) pileus covering.

Etymology: from *purpureus* (L) purple, and *squamulosus* (L) with small scales, as the basidiomata are covered with purple squamules.

Diagnosis: *Xanthagaricus purpureosquamulosus* has slender basidiomata, characterized by a pileus with brownish-grey to violet-brown squamules on a pale-violet to violet background, brownish-grey to greyish-brown appendiculate velar remnants, white to greyish lamellae, a stipe with greyish-yellow to light-brown squamules, and a small apical annulus, ellipsoid-ovoid basidiospores, which are convex on adaxial side, with hyaline to a pale-brown wall, short clavate to ellipsoid cheilocystidia, an irregular epithelium as pileus and stipe covering, and no clamp connections.

Holotype: MFLU 19-2354.

Pileus 4.5–10 mm, first paraboloid or hemispherical, expanding to convex, plano-convex, with or without low umbo, when young completely covered with crowded squamules, brownish-grey (9D2–3) to violet-brown (10D4–8), soon breaking up into greyish-brown (9E2, 9D3) to violet-brown (10E4) squamules, crowded around center or umbo and with radially arranged squamules toward the margin, on pale-violet to violet (17A3–7) background and paler when mature; margin incurved, with brownish-grey (9D2–3) to greyish-brown (9E2, 9D3) appendiculate triangular velar remnants. Lamellae, l = 3–4 series, free, 1.5–2 mm wide, white to yellowish-white (4A2), becoming yellowish-grey to greyish-yellow (4B2–3) when mature, broadly ventricose, with white, smooth to slightly eroded margin. Stipe 15–30 × 1–1.5 mm, slender and tapering downwards, covered with white to yellowish-white (4A2) squamules at apex, bellow annulus with crowded squamules toward the base, greyish-yellow, greyish-brown to light-brown (6D3–5), on yellowish-white to pale-yellow (4A2–3) background, darker at base zone, with greyish-brown (7E3–4) squamules. Annulus attached to apical zone of stipe, with velar remnants and concolorous to those on pileus margin, sometimes fragile with age. Context white in pileus, up to 1 mm thick at center; hollow in stipe and concolorous with surface. Odor and taste not observed. Spore pint greyish-orange (5B3).

Basidiospores [50,2,2] 4.5–6 × 3.5–4 μm, avl × avw = 5.34 × 3.78 μm, Q = 1.3–1.5, avQ = 1.4, ellipsoid-ovoid in frontal view, ellipsoid in side-view, thick-walled, smooth, hyaline to pale-brown, without a germ pore. Basidia 10–12 × 5.5–7 μm, obovoid, short clavate to ellipsoid, thin-walled, hyaline, 4-spored. Pleurocystidia absent. Cheilocystidia 10–16 × 5–8 μm, short clavate to clavate, ellipsoid, some branched, slightly thick-walled, hyaline. Pileus covering an irregular epithelium composed of globose to subglobose (7.5–25 μm wide) cells in the upper layer, broadly ellipsoid to oblong cells (10–62 × 5–15 μm) in the lower layer, thin-walled, smooth, with pale-brown to brown parietal and intracellular pigments, with pale-brown to hyaline, up to 7 μm wide hyphae at base of the epithelial layers. Stipe covering an irregular epithelium same as on pileus. Clamp connections absent.

Habitat and distribution: growing in small to large groups, saprotrophic on humus-rich soil with dead leaves and wood under trees of *Samanea saman* or beside grassland; commonly found on Mae Fah Luang University Campus, Chiang Rai, Thailand.

Additional material examined: Thailand, Chiang Rai Province, Muang District, Mae Fah Luang University Campus, 26 October 2018, P. Sysouphanthong, PS2018-217 (MFL19-2356, paratype).

Notes: there are only two species of *Xanthagaricus*, which resemble *X. purpureosquamulosus* in pileus color. *X. caeruleus* Iqbal Hosen, T.H. Li and Z.P. Song, described from China has pale-lilac to grayish-lilac or grayish-violet squamules on a pale-grayish-lilac to violet white background, but it differs from *X. purpureosquamulosus* by having white lamellae, which turn light-blue or pastel blue to grayish and finally ink-blue or blackish blue [74]. Based on the phylogenetic analysis of ITS and LSU sequence data, *X. caeruleus* is not closely related to *X. purpureosquamulosus* (Figure 1). The second species with similar colors is *X. ianthinus* Y. Li and F.J. Wang, also from China, which differs from *X. purpureosquamulosus* by having bluish-violet to violet, more or less violet-brown squamules, white velar remnants, and yellowish-white to light-pinkish-white (4A2) lamellae [75]. It is not close to

*X. purpureosquamulosus* either in the phylogenetic tree based on ITS and LSU sequences (Figure 1).

*Xanthagaricus thailandensis* J. Kumla, N. Suwannarach, and S. Lumyong, recently described from Thailand, differs in the larger pileus (30–45 mm), and the pale-orange to grayish-orange, large squamules on the pileus [35].

The phylogenetic analysis of the ITS and LSU sequences (Figure 1) shows that *X. purpureosquamulosus* is closely related to a group of species with yellow to olive-brown squamules, viz. *X. epipastus* (Berk. and Broome) S. Hussain, known from Sri Lanka [33,76], *X. pakistanicus* Hussain, Afshan and Ahmad from Pakistan [77], and *X. necopinatus*.

**Author Contributions:** Conceptualization, P.S., E.C.V., N.T., and J.-K.L.; methodology, E.C.V., P.S., J.-K.L., and N.T.; formal analysis, J.-K.L., E.C.V., P.S., and N.T.; data curation, E.C.V., P.S., N.T., and J.-K.L.; supervision, N.T.; and E.C.V.; writing—original draft, P.S., E.C.V., N.T., and J.-K.L.; writing—review and editing, E.C.V., P.S., N.T., and J.-K.L. All authors have read and agreed to the published version of the manuscript.

**Funding:** we were supported by Thailand research fund grants "Study of saprobic Agaricales in Thailand to find new industrial mushroom products" (Grant No. DBG6180015) and Thailand Science Research and Innovation (TSRI) grant, Macrofungi diversity research from the Lancang-Mekong Watershed and surrounding areas (Grant No. DBG6280009). The fieldwork of ECV in 2007 was conducted under an MTA agreement with Dr. Kasem Soytong of KMITL. Financial support of ECV and PS was from NSF grant DEB 0618293.

**Institutional Review Board Statement:** Not applicable.

**Informed Consent Statement:** Not applicable.

**Data Availability Statement:** Data can be found within the manuscript.

**Acknowledgments:** Kevin D. Hyde is thanked for his advising and providing a laboratory for fieldwork at the Mushroom Research Centre in Chiang Mai, Thailand. Mae Fah Luang University is thanked for providing laboratory facilities for microscopical study and herbarium. Biotechnology and Ecology Institute, Ministry of Science and Technology of Laos is also thanked for depositing herbarium.

**Conflicts of Interest:** The authors declare no conflict of interest.

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
