# Peer review of "Description of Lepiotaceous Fungal Species of the Genera Chlorophyllum, Clarkeinda, Macrolepiota, Pseudolepiota, and Xanthagaricus, from Laos and Thailand"

_diversity, doi:10.3390/d13120666_

Round 1
Reviewer 1 Report
Some lepiotaceous species from Laos and Thailand: Chlorophyllum, Clarkeinda, Macrolepiota, Pseudolepiota, and Xanthagaricus Phongeun
The author's collected fungal species from Laos and Thailand, provide phylogenetic placement, descriptions, and color images, along with two new species descriptions.
The authors have completed a lot of work and the manuscript is organized and easy to follow.
There are only three minor changes along with some uneven spacing.
Line 117: H2O... the O is a zero (0) and not an O. Appears out of place, please replace with an O
line 120: First words of a sentence should be spelled out. Please spell out 25.
line 141: Vellinga is outgroup. This sentence is awkward as written.
Author Response
Response to Reviewer 1 Comments
We would like to thank to the reviewer to have comments and correcting our article. We have corrected comments of the reviewer as below. And the yellow marks in the manuscript are what we corrected all reviewers’ comments.
- Line 117: It has been changed from “(H20)” to be “(H2O)”
- Line 120: It has been changed from “25” to be “Twenty-five”
- Line 141: It has been changed from “Chlorophyllum rhacodes (Vittad.) Vellinga is outgroup” to be “Chlorophyllum rhacodes (Vittad.) Vellinga is set as outgroup”.

Reviewer 2 Report
I reviewed the manuscript entitled “Some lepiotaceous species from Laos and Thailand: Chlorophyllum, Clarkeinda, Macrolepiota, Pseudolepiota, and Xanthagaricus” submitted by P. Sysouphanthong, N. Thongklang, J-K. Liu and E.C. Vellinga to Diversity journal.
The authors are making a detailed description of 12 fungal species, whose two species were unknown and are new, two species were recorded in a new geographic area, and eight species were re-described in details after molecular identifications, not yet performed before, that confirm their taxonomy.
This kind of paper is very specific and intended to mycologists mainly, but I think that such studies are an essential step in order to improve our knowledge about fungal diversity, and to allow the proper identification of fungi.
The paper is well written and well structured. The characteristics of described fungal species are really detailed and clearly explained. However, some important confusions in the Figures and Figure legends need to be corrected.
I would suggest the following revisions:
- Title: I think that the title is not informative enough; moreover, it is mentioned: “Some lepiotaceous species[…]:” but some genera are cited and not species. I would suggest to edit the title with, for example, “Description of lepiotaceous fungal species from Laos and Thailand, belonging to five genera: Chlorophyllum, Clarkeinda, Macrolepiota, Pseudolepiota, and Xanthagaricus”, or “Description of lepiotaceous fungal species of the genera Chlorophyllum, Clarkeinda, Macrolepiota, Pseudolepiota, and Xanthagaricus, from Laos and Thailand”, …
- l.112: Do you mean “Laos specimens” instead of “Lao specimens”?
- l.213: I suggest starting a new paragraph from “Figure 4 shows […]”, as you did above for Figure 1, 2 and 3.
- Figure 12, 16, 18: I do not see on the Figure the “(d)” mentioned in the legend.
- Figure 12: From your text it should correspond to Chlorophyllum molybdites and not Chlorophyllum hortense, which is already presented in Figure 10. Please correct this.
- Figure 16: From your text it should correspond to Macrolepiota detersa and not Clarkeinda trachodes, which is already presented in Figure 14. Please correct this.
- Figure 22: From your text it should correspond to Macrolepiota velosa and not Macrolepiota excelsa, which is already presented in Figure 20. Please correct this.
- l.761: “Macrolepiota excelsa is a beautiful species” Please, edit this sentence in order to remove the term “beautiful” which is too subjective and not scientific.
- Figures 18, 20 and 22: These 3 Figures are exactly the same, maybe it is due to the confusions in the Figures/legends? But, in the case where some Figures are the same, so could you please merge the legends and remove the replicated Figures? Also, please check that the scales on the Figures correspond to the size ranges described in the Results.
- Figures 26 and 28: These two Figures are identical, so I suggest to merge the legends and to remove one Figure. This repetition of identical Figures makes the manuscript unnecessarily long.
Author Response
Response to Reviewer 2 Comments
We would like to thank you very much for your kindly check in all detail of our article. We did wrong arranging photos and legends during formatting. We followed your comments and details of correcting are below. And the yellow marks in the manuscript are what we corrected all reviewers’ comments.
- Topic: We agree to have the topic “Description of lepiotaceous fungal species of the genera Chlorophyllum, Clarkeinda, Macrolepiota, Pseudolepiota, and Xanthagaricus, from Laos and Thailand”.
- Line 112:
- Question of review: Do you mean “Laos specimens” instead of “Lao specimens”?
- Response of authors: We would like to say that Lao specimens are specimens come from Laos country (Laos= noun, Lao = adjective).
- Line 213:
- comment of review: I suggest starting a new paragraph from “Figure 4 shows […]”, as you did above for Figure 1, 2 and 3.
- Response of authors: We have reformatted and start new
- Figure 12: the numbers has been rearranged.
- Figure 16: the numbers has been rearranged.
- Figure 18: the numbers has been rearranged.
- Figure 12: the name “Chlorophyllum hortense” has been chaged to be “Chlorophyllum molybdites”.
- Figure 16: the name “Clarkeinda trachodes” has been changed to be “Macrolepiota detersa”.
- Figure 22: the name “Macrolepiota excelsa” has been changed to be “Macrolepiota velosa”.
- Line 761: the word “beautiful” has been changed to be “new”.
- Figure 20: the old photo has been replaced by new photo.
- Figure 22: the old photo has been replaced by new photo.
- Figures 28: the old photo has been replaced by new photo.

Reviewer 3 Report
The manuscript “Some lepiotaceous species from Laos and Thailand: Chlorophyllum, Clarkeinda, Macrolepiota, Pseudolepiota, and Xanthagaricus” by Sysouphanthong et al. provides an taxonomic overview over a number of lepiotaceous species in the genera indicated in the title. Specimens of fruiting bodies were collected in various habitats in Laos and Thailand. Macroscopic and microscopic morphology was documented by photographs, drawings and biometric measurements. Sequences from the ITS, LSU and rpb2 region were generated and deposited into GenBank. Phylogenetic trees were calculated based on maximum likelihood and Bayesian inferences. Based on phylogenetic trees and morphological characters, the manuscript describes two new species: Xanthagaricus purpureosquamulosus and Macrolepiota excels. Moreover, the manuscript contains a number of first records for the investigated area. The descriptions of the species are accompanied by additional information from literature regarding distribution of the species, habitat, etc.
The manuscript is well written and structured. As far as I (a forest pathologist with no experience with these fungal taxa) can tell, it is based on very thorough and sound work.
The manuscript contains many self-citations, but this may be justified considering the number of papers that the authors have published in this particular field/topic. I believe that the manuscript provides valuable information to the mycological community.
I inserted few specific comments of minor importance in the attached pdf, of which I hope that they may help improving the manuscript.

Author Response
Response to Reviewer 3 Comments
Many thanks for reviewer’s comments, please kindly see our correcting as below. And the yellow marks in the manuscript are what we corrected all reviewers’ comments.
- Line 60: It has been changed from “The occurrence of rhacodes has not been confirmed” to be “The occurrence of Chl. rhacodes has not been confirmed” in Thailand”.
- Line 61: It has been changed from “but here we report globosum (Mossebo) Vellinga with green basidiospores” to be “but in this article, we report Chl. globosum (Mossebo) Vellinga with green basidiospores”.
- Line 105: It has been changed from “the ecology and habitat of mushrooms were recorded” to be “the forest type and habitat or substrate of mushrooms were recorded”
- Line 123: The word “quotient” was deleted.
- Line 138:
- Question 1 from reviewer: Why only 2 LSU? Were the sequences of other specimens of bad quality? If so, it may be an interesting information for the reader how many sequences per marker were successfully generated, and it would make the compilation of the datasets appear less arbitrary.
- Response of authors: We only have two new LSU sequences for the new species “Xanthagaricus purpureosquamulosus”. For our other species in this dataset, we only have ITS sequences. However, other representative sequences in the dataset can be used for combined markers.
- Question 2 from reviewer: These were selected by applying which criteria?
- Response of authors: We selected the sequences and species according to relation of genera and species, and representatives of genera and species in the family Agaricaceae.
- Line 484:
- Comment from reviewer: This paper may be of interest here: Wang, N., Zhao, Z., Gao, J., Tian, E., Yu, W., Li, H., ... & Chen, A. (2021). Rapid and Visual Identification of Chlorophyllum molybdites With Loop-Mediated Isothermal Amplification Method. Frontiers in Microbiology, 12, 407.
- Response of authors: Thank you very much for the article from reviewer, the paper is new publishing and interesting for recognizing Chlorophyllum molybdites, and this is new method to identify mushrooms quickly.
- Line 504:
- Comment from reviewer: It feels as if there should be a reference for this statement.
- Response of authors: Two references about the species and text have been added.
- [76] Bijeesh, C., K. B. Vrinda, and C. K. Pradeep. "Mushroom poisoning by Chlorophyllum molybdites in Kerala." Mycopathol. Res54.4 (2017): 477-483.
- [77] Wang, N., Zhao, Z., Gao, J., Tian, E., Yu, W., Li, H., ... & Chen, A. (2021). Rapid and Visual Identification of Chlorophyllum molybdites With Loop-Mediated Isothermal Amplification Method. Frontiers in Microbiology, 12, 407.
